# CI-CBM: Class-Incremental Concept Bottleneck Model for Interpretable Continual Learning

**Amirhosein Javadi**                                           *amjavadi@ucsd.edu*
*Department of Electrical and Computer Engineering, University of California San Diego*

**Tuomas Oikarinen**                                           *toikarinen@ucsd.edu*
*Department of Computer Science and Engineering, University of California San Diego*

**Tara Javidi**                                           *tjavidi@ucsd.edu*
*Department of Electrical and Computer Engineering, University of California San Diego*

**Tsui-Wei Weng**                                           *lweng@ucsd.edu*
*Halıcıoğlu Data Science Institute, University of California San Diego*

**Reviewed on OpenReview:** *https://openreview.net/forum?id=Wf6OpLgj2i*

## Abstract

Catastrophic forgetting remains a fundamental challenge in continual learning, in which models often forget previous knowledge when fine-tuned on a new task. This issue is especially pronounced in class incremental learning (CIL), which is the most challenging setting in continual learning. Existing methods to address catastrophic forgetting often sacrifice either model interpretability or accuracy. To address this challenge, we introduce Class-Incremental Concept Bottleneck Model (**CI-CBM**), which leverage effective techniques, including concept regularization and pseudo-concept generation to maintain interpretable decision processes throughout incremental learning phases. Through extensive evaluation on seven datasets, **CI-CBM** achieves comparable performance to black-box models and outperforms previous interpretable approaches in CIL, with an average 36% accuracy gain. **CI-CBM** provides interpretable decisions on individual inputs and understandable global decision rules, as shown in our experiments, thereby demonstrating that human-understandable concepts can be maintained during incremental learning without compromising model performance. Our approach is effective in both pretrained and non-pretrained scenarios; in the latter, the backbone is trained from scratch during the first learning phase. Code is publicly available at github.com/importAmir/CI-CBM.

## 1 Introduction

Deep learning models have demonstrated exceptional performance when trained on large-scale, stationary datasets all at once, as evidenced by breakthroughs in computer vision (He et al., 2016), healthcare (Ronneberger et al., 2015; Zhang et al., 2024), and robotics (Mnih, 2013). However, their performance deteriorates substantially when data arrives sequentially over time (De Lange et al., 2021). Continuously retraining models from scratch each time new data arrives is computationally expensive, time-consuming, and impractical. These limitations have spurred growing interest in continual learning, a different learning paradigm that seeks to enable models to learn from evolving data streams without forgetting previously acquired knowledge.

Continual learning (Parisi et al., 2019) enables models to incrementally update their knowledge and adapt to new data over time. In this work, we focus on the class incremental learning (CIL) setting (Van de Ven & Tolias, 2019), which is widely regarded as the most challenging form of continual learning. In CIL, each phase introduces a new set of classes disjoint from previously seen ones, and at inference time, a single model must classify test samples from all observed classes without access to phase identifiers. A central difficulty

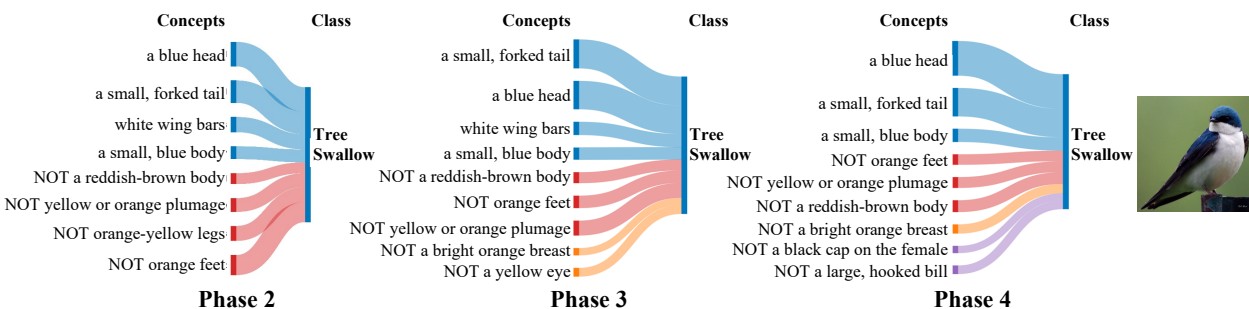

Figure 1: Visualization of the final layer weights with absolute values greater than 0.2 for the **Tree Swallow** class in the CUB dataset under a four-phase scenario. Concepts with negative weights are labeled as "NOT" concepts. Positive and negative concepts in phase 2 are shown in blue and red, respectively, while concepts added in phases 3 and 4 are shown in orange and purple. As new phases arrive, CI-CBM can preserve the positive concepts while learning more discriminative negative features. The thickness of each edge corresponds to the absolute value of the weight. Additional visualizations are provided in Section A16.

in this setting is *catastrophic forgetting* (Goodfellow et al., 2013), where learning new tasks often interferes with previously acquired knowledge, leading to substantial accuracy degradation on earlier tasks.

To address catastrophic forgetting in CIL, conventional methods utilize a bounded buffer to store a subset of exemplars (Rebuffi et al., 2017; Hou et al., 2019; Wu et al., 2019) from previous phases. The model is then fine-tuned jointly on both the current phase data and the stored data. However, this approach raises concerns regarding privacy and storage limitations. Recently, the Exemplar-Free CIL (EFCIL) approach has gained increasing attention (Zhu et al., 2021b; 2022; Petit et al., 2023). The main challenge in this approach is to classify between old and new classes without access to old data. Some approaches (Zhu et al., 2021b; 2022) propose fine-tuning the model on new classes while employing knowledge distillation to preserve the learned knowledge from old classes. Others (Petit et al., 2023; Panos et al., 2023) freeze the feature extractor after the first phase and focus on incrementally learning the classifier in the next phases. Other methods (Wang et al., 2022c;b; Smith et al., 2023) employ a backbone model that has been pretrained on large-scale datasets. However, these models depend heavily on strong pretraining and exhibit a significant performance drop when the backbone is trained on first-phase data instead of a large-scale dataset (Tang et al., 2023).

Although recent advancements in CIL offer promising solutions to catastrophic forgetting, the decision-making mechanisms of these models are often difficult to understand and are regarded as *black-box* processes. Interpretability is crucial for uncovering the information a model uses when classifying inputs, which is critical for identifying biases. Researchers have developed methods to interpret black-box deep neural networks (DNNs) in continual learning (Patra & Noble, 2020). However, most methods focus on examining the DNN model after training is complete, with only a few approaches, such as ICICLE (Rymarczyk et al., 2023), IN2 (Yang et al., 2024), CONCIL (Lai et al., 2025), and CLG-CBM (Yu et al., 2025), focusing on learning models that are interpretable by design. Despite being pioneering work in this direction, ICICLE is restricted to specific architectures and fine-grained datasets, while IN2's suboptimal concept expansion compromises efficiency, and CONCIL relies on per-sample concept annotations. CLG-CBM and the other mentioned interpretable CIL methods rely on pretrained backbones, which can blur the CIL objective of learning beyond what is already encoded in pretraining. To our knowledge, our proposed method is the first interpretable CIL model to demonstrate robustness in both pretrained and non-pretrained settings. Moreover, the performance gap between current interpretable models and unrestricted ones in CIL further diminishes the incentive to adopt interpretable approaches, as their reduced accuracy makes the models less practical for real-world use. These limitations motivate our work to develop a more generalizable and inherently interpretable approach to class incremental learning, suitable for both pretrained and non-pretrained model scenarios.

In this paper, we propose a new an effective framework called the Class Incremental Concept Bottleneck Model (**CI-CBM**) to extend the Concept Bottleneck Model (CBM) for the challenging Exemplar-Free CIL (EFCIL) setting. In the EFCIL setting, due to the absence of samples from previous classes, the learned functionality of each concept and its contribution to those classes must be preserved while allowing the model to continually learn new concepts and adjust their contributions across all classes. To address these

problems, **CI-CBM** employs an effective mechanism to prevent concept drift and mitigate classifier bias (see Figure 1 and 6). Our contributions are summarized below:

- We introduce a new approach to learn inherently interpretable neural models for class incremental learning with much better performance and utility. Unlike previous interpretable approaches in CIL, our approach **CI-CBM** achieves 25–43% higher accuracy across multiple benchmarks while maintaining scalability and adaptability across various datasets and architectures, demonstrating strong performance both with pretrained backbone models and when trained from scratch.

- We propose new techniques including concept regularization to prevent the learned concepts from losing their functionality while learning new concepts. Additionally, we utilize pseudo-concepts for previous class data in conjunction with actual concepts for new class data to incrementally train the sparse classifier across different phases.

- We evaluate our approach by conducting comprehensive EFCIL evaluation scenarios and performing ablation studies to assess the impact of different components of the proposed method. The results demonstrate the superiority of our method over other interpretable models in CIL, achieving an average accuracy gain of 36%.

## 2 Related work

### 2.1 Class Incremental Learning

Class Incremental Learning (CIL) aims to extend the capabilities of deep learning models to continuously learn from new data and adapt to new classes over time while retaining previously acquired knowledge. The goal is to develop a model that can classify all previously seen classes effectively at any stage of training, which requires a balance between plasticity, the flexibility to learn new features, and stability, the resistance to forgetting old information. The phenomenon of catastrophic forgetting (Goodfellow et al., 2013) illustrates a fundamental trade-off in CIL, where the effort to incorporate new information can result in the loss of valuable knowledge from previous phases, leading to a sharp drop in performance on previous classes.

To address catastrophic forgetting, several approaches have been proposed: **(I)** Regularization-based methods: These approaches (Kirkpatrick et al., 2017; Wang et al., 2021) mitigate forgetting by constraining changes in critical parameters across phases. However, these methods lack reliable metrics for parameter importance and perform poorly in CIL (Van de Ven & Tolias, 2019). **(II)** Architecture-based methods: These approaches expand network capacity dynamically when a new phase arrives (Rusu et al., 2016; Yan et al., 2021). However, a key challenge is the increasing memory and computational costs as the architecture grows, making it crucial to manage the rate of expansion with each phase. **(III)** Rehearsal-based methods: These approaches utilize various sampling strategies, such as herding (Rebuffi et al., 2017), diversity-aware sampling (Bang et al., 2021), reservoir sampling (Buzzega et al., 2020), and greedy sampling (Prabhu et al., 2020), to store samples in bounded memory. These samples are then used for knowledge distillation (Li & Hoiem, 2017; Rebuffi et al., 2017; Dhar et al., 2019), bias correction (Hou et al., 2019; Wu et al., 2019), or gradient regularization (Lopez-Paz & Ranzato, 2017; Chaudhry et al., 2018). Despite their effectiveness in addressing catastrophic forgetting, rehearsal-based methods pose privacy risks, as storing data from previous phases may expose confidential information. Generative models can be trained to produce samples from previous classes (Wu et al., 2018; Gao & Liu, 2023), but they are prone to catastrophic forgetting (Thanh-Tung & Tran, 2020) and are vulnerable to model-inversion attacks (Zhang et al., 2020).

Recently, Exemplar-Free Class Incremental Learning (EFCIL) approaches have gained popularity. These methods often focus on learning high-quality feature representations in the first phase, which usually covers around half of the total classes. For instance, FeTrIL (Petit et al., 2023) freezes the feature extractor after this initial phase and generates pseudo-features using basic geometric transformations based on the class mean. The aim of such methods is to maximize the utility of the representations learned in the first phase for the subsequent phases. FeTrIL++ (Hogea et al., 2024) extends FeTrIL by additionally storing the per-class diagonal covariance alongside each class centroid, and by applying a lightweight optimization to align the variance of pseudo-features with that of the original features.

Recent advancements have introduced methods that leverage ImageNet-pretrained ViT models. Among these, some approaches (Wang et al., 2022c;b;a; Tang et al., 2023) introduce lightweight prompts that are concatenated with input patches and processed alongside them, enabling task-specific adaptation without modifying backbone weights. These methods maintain a pool of prompts, selecting or generating instance-specific ones through key-query matching (Wang et al., 2022c), clustering (Wang et al., 2022a), or attention-based combinations (Smith et al., 2023). However, these methods require a strong pretrained backbone and tend to experience performance degradation when the data from the first phase is used for pretraining the backbone (Tang et al., 2023). In addition, the core idea of CIL is to enable a system to acquire knowledge that was previously unavailable (Zhou et al., 2024b). The use of large pretraining datasets like ImageNet raises the question of whether these models encounter truly novel information. However, our approach reduces dependence on extensive pretraining, making it applicable to both pretrained and non-pretrained scenarios.

## 2.2 Interpretability

Despite advancements in CIL methods, there remains a gap in understanding how these black-box models function. Interpretability frameworks aim to make the decision-making process of models more transparent. Rather than relying on post-hoc explanations for black-box models, it is more effective to design models that are inherently interpretable, reducing the risk of misleading interpretations (Rudin, 2019).

The Concept Bottleneck Model (CBM) (Koh et al., 2020) incorporates an intermediate concept bottleneck layer, where each neuron corresponds to a human-understandable concept. This allows the model's final prediction to be expressed as a linear combination of interpretable concepts, significantly improving our insight into how decisions are made. CBM requires dense concept annotations in the training data to learn the bottleneck layer, limiting its scalability and applicability. LF-CBM (Oikarinen et al., 2023) addresses this issue by automating the CBM training process, thus reducing reliance on human experts. LF-CBM leverages Large Language Models, such as GPT-3, to gather relevant concepts for each task and aligns image-concept activations using vision-language models like CLIP (Radford et al., 2021). CBMs have demonstrated significant potential across domains, including medical applications (Yuksekgonul et al., 2022), deep generative models (Ismail et al., 2023; Kulkarni et al., 2025), and large language models (Sun et al., 2025).

Although interpretability is well-established in classical machine learning, it remains underexplored in the context of CIL. ICICLE (Rymarczyk et al., 2023) introduced interpretability in class incremental learning through prototypical parts. However, their work is limited to specific model architectures and fine-grained datasets. IN2 (Yang et al., 2024) extended the CBM by freezing learned concepts and regularizing the prediction layer. However, the expansion of the concept set in their approach is not optimal, and its performance is much weaker than that of unrestricted models. CONCIL (Lai et al., 2025) employed random-feature expansion and performed continual concept prediction and classification via closed-form ridge-regression updates; however, it relies on a pretrained backbone and per-sample concept annotations, limiting practicality in many continual learning settings. Most similar to our approach is CLG-CBM (Yu et al., 2025), which leverages a frozen, large-scale pretrained CLIP backbone and synthesizes pseudo-features for prior classes via prototype-based augmentation. However, this reliance on strong pretrained backbone may weaken the continual learning premise of acquiring genuinely new knowledge beyond what is already encoded in pretraining.

Generally, there is a trade-off between interpretability and accuracy. Previous works, such as PCBM (Yuksekgonul et al., 2022) (their Tables 1 and 2) demonstrate that interpretable models generally suffer from lower accuracy compared to their black-box counterparts in classical machine learning. LF-CBM (Oikarinen et al., 2023) (their Table 2) alleviates this trade-off by improving accuracy in standard image classification tasks. We found that the gap is more pronounced in the context of CIL. As the performance gap between interpretable and unrestricted models widens, the motivation to use interpretable models diminishes. In contrast, our proposed method bridges this gap by providing interpretability with minimal accuracy gap.

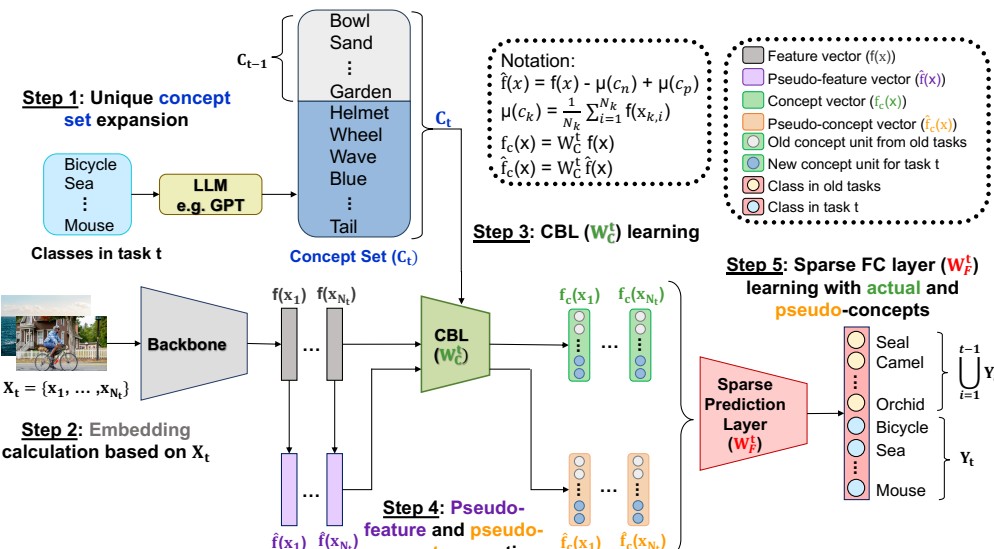

Figure 2: Overview of our pipeline for Class Incremental Concept Bottleneck Model (CI-CBM). Color-coded text matches the corresponding stages in the figure.

## 3 Methods

To make this work self-contained, we begin by reviewing the Label-free Concept Bottleneck Models (LF-CBM) framework and then explain how we adapt it for the EFCIL scenario.

### 3.1 Label-free Concept Bottleneck Models (LF-CBM)

**Concept Set Creation and Filtering.** The concept set refers to the group of concepts included in the interpretable concept bottleneck layer. These concepts represent features that are both important to the problem and easy for humans to understand, such as key features, common surrounding items, and superclasses. These concepts are generated automatically by prompting GPT-3 to identify concepts related to each class in the dataset. The concepts for all classes are then combined to form an initial concept set. This set is further refined by filtering out concepts that are too lengthy, those that closely resemble the class names, and concepts that are similar to each other.

**Learning the Concept Bottleneck Layer.** The next step involves learning the projection weights $W_C$, which map the frozen backbone model's feature space onto a space where the axes correspond to interpretable concepts. Let $C = \{t_1, \ldots, t_M\}$ represent the concept set, and $D = \{x_1, \ldots, x_N\}$ the training dataset. The concept activation matrix $P$ is calculated and stored, where $P_{i,j} = E_I(x_i) \cdot E_T(t_j)$, with $E_I$ and $E_T$ denoting the CLIP image and text encoders, respectively. The weights $W_C$ are optimized to maximize the similarity between the neuron's activation patterns and the target concepts. The similarity is measured using the differentiable *cosine-cubed* function, defined as:

$$L(W_C) = \sum_{i=1}^{M} -\text{sim}(t_i, q_i) := \sum_{i=1}^{M} -\frac{\bar{q}_i^3 \cdot \bar{P}_{:,i}^3}{\|\bar{q}_i^3\|_2 \|\bar{P}_{:,i}^3\|_2} \tag{1}$$

Here, $q_i$ denotes the activation of the $i$-th neuron in the projection layer, and $\bar{q}$ represents $q$ normalized to have mean 0 and standard deviation 1.

**Learning the Sparse Final Layer.** The final step involves learning a sparse predictor, $(W_F, b_F)$, using the GLM-SAGA (Wong et al., 2021) solver with an elastic net objective.

$$\min_{W_F, b_F} \sum_{i=1}^{N} L_{ce}(W_F f_c(x_i) + b_F, y_i) + \lambda R_\alpha(W_F) \tag{2}$$

---

**Algorithm 1** CI-CBM Training in Exemplar-Free Class-Incremental Learning

---

**Require:** Phase datasets $\{D_t = (X_t, Y_t)\}_{t=1}^T$; backbone $f$; initial concept set $C_1$; distillation weight $\beta$.
 1: Initialize phase $t = 1$ concept set $C_1$ via LLM prompts and filtering.
 2: Compute $P_1$ using a vision-language model: $P_1[i,j] \leftarrow E_I(x_i)^\top E_T(c_j)$ for $x_i \in X_1, c_j \in C_1$.
 3: Learn $W_C^1$ by minimizing concept alignment loss (Eq. 1).
 4: Train sparse predictor $(W_F^1, b_F^1)$ on concept features $f_c(x) = W_C^1 f(x)$ (Eq. 2).
 5: Store class centroids $\mu(c)$ for all classes seen in phase 1 in backbone feature space.
 6: **for** $t = 2$ **to** $T$ **do**
 7:    **(Module I) Concept set expansion:**
 8:    Generate candidate concepts for new classes in $Y_t$ using LLM prompts.
 9:    Using cosine similarity in text-embedding space, remove candidates that are too similar to any seen
      class name or near-duplicates of existing concepts in $C_{t-1}$; remove lengthy concepts.
10:    Update concept set $C_t \leftarrow C_{t-1} \cup \{\text{filtered new concepts}\}$.
11:    Compute $P^t$ on current data: $P^t[i,j] \leftarrow E_I(x_i)^\top E_T(c_j)$ for $x_i \in X_t, c_j \in C_t$.
12:    **(Module II) Concept bottleneck learning with distillation:**
13:    Cache previous outputs on current data for old concepts: $q^{t-1}(x) \leftarrow W_C^{t-1} f(x)$ for $x \in X_t$.
14:    Expand $W_C^{t-1}$ with new neurons to match $|C_t|$.
15:    Learn $W_C^t$ by minimizing alignment + distillation loss (Eq. 3) with weight $\beta$.
16:    **(Module III) Pseudo-feature and pseudo-concept generation:**
17:    Compute and store new class centroids $\mu(c)$ for classes in $Y_t$.
18:    **for** each past class $c_p$ **do**
19:      Find nearest new class $c_n$ by cosine similarity between centroids.
20:      Generate pseudo-features: $\hat{f}(c_p) \leftarrow f(c_n) - \mu(c_n) + \mu(c_p)$                      (Eq. 4)
21:      Project to pseudo-concepts: $\hat{f}_c(c_p) \leftarrow W_C^t \hat{f}(c_p)$                              (Eq. 5)
22:    **end for**
23:    Train $(W_F^t, b_F^t)$ using real concepts for new classes and pseudo-concepts for past classes (Eq. 6).
24: **end for**

---

where $R_\alpha(W_F) = \frac{1-\alpha}{2}\|W_F\|_F^2 + \alpha\|W_F\|_{1,1}$., with $\|\cdot\|_F$ representing the Frobenius norm and $\|\cdot\|_{1,1}$ the element-wise matrix norm, and $f_c(x) = W_C f(x)$ denotes the projection of backbone features into the concept space.

## 3.2 Class Incremental Concept Bottleneck Model (CI-CBM)

To adapt LF-CBM for the class incremental learning setting, we introduce three main modules, as illustrated in Figure 2: (I) curated concept set expansion (Step 1), (II) knowledge-preserving concept learning (Step 2-3), and (III) dynamic adaptation in the sparse prediction layer (Step 4-5). Let $D = \{D_1, D_2, \ldots, D_T\}$ denote the sequence of datasets from the first to the last phase, where each dataset $D_t = \{X_t, Y_t\} = \{x_{t,j}, y_{t,j}\}_{j=1}^{N_t}$ consists of $N_t$ labeled samples received by the model at phase $t$. The classes across different phases are disjoint, and the phase boundaries remain unknown. When the first batch of data, $D_1$, arrives, the model is trained following the LF-CBM approach, as outlined in Section 3.1. After completing the learning process for phase $t-1$, the model has a concept set $C_{t-1}$, containing $M_{t-1}$ concepts related to the previously seen classes $(\bigcup_{i=1}^{t-1} Y_i)$, along with a concept bottleneck layer $W_C^{t-1}$ and a unified sparse prediction layer $(W_F^{t-1}, b_F^{t-1})$.

**Module (I): Incremental Concept Set Expansion.** When a new phase $t$ begins, concepts for the new classes are generated using GPT-3, filtered, and added to the concept set. To avoid adding redundant concepts during expansion, we embed each candidate concept, each existing concept, and each class name of seen classes using a text encoder (Sentence-Transformer and the CLIP text encoder) and compute cosine similarity. We discard a candidate if its similarity exceeds 0.9 to any existing concept or exceeds 0.85 to any class name. Full generation/filtering pipeline details are provided in Appendix A11. As a result, the updated concept set $C_t$ contains $M_t$ unique concepts. Next, the concept activation matrix $P^t \in \mathbb{R}^{N_t \times M_t}$ is computed using a multimodal model based on the new training dataset $D_t$ and the combined concept set $C_t$. In our experiments, we used SigLIP (Zhai et al., 2023) instead of CLIP (Radford et al., 2021), as the

multimodal model for computing $P^t$. SigLIP is a recent model that focuses on image-text pairs and employs a sigmoid loss function, in contrast to the softmax-based contrastive learning approach used in CLIP.

**Module (II): Preventing Concept Drift in the Concept Bottleneck Layer with Distillation.** As the concept set expands, the learned $W_C^{t-1}$ is adjusted to accommodate new neurons for the added concepts, resulting in $W_C^t$, which must be learned. Naively fine-tuning $W_C^t$ on $D_t$ risks shifting the functionality of previously learned concepts to the new data, leading to catastrophic forgetting of past knowledge. Conversely, freezing the learned concepts to prevent updates limits the model's adaptability. Before expanding $W_C^{t-1}$ to accommodate new concepts, the scores for the current concepts on the dataset from the new phase are calculated and saved. During the training of the concept bottleneck layer, an additional distillation loss is introduced to regularize the loss function, aiming to prevent the output of the current model ($W_C^t$) from drifting too far from the saved output of the previous model ($W_C^{t-1}$). The loss function is defined as:

$$L(W_C^t) = \sum_{i=1}^{M_t} -\frac{\overline{q_i^t}^3 \cdot \overline{P_{:,i}^t}^3}{\|\overline{q_i^t}^3\|_2 \|\overline{P_{:,i}^t}^3\|_2} + \beta \sum_{i=1}^{M_{t-1}} -\frac{\overline{q_i^t}^3 \cdot \overline{q_i^{t-1}}^3}{\|\overline{q_i^t}^3\|_2 \|\overline{q_i^{t-1}}^3\|_2} \tag{3}$$

where $\beta \geqslant 0$ is a scalar hyperparameter that weights the distillation regularizer.

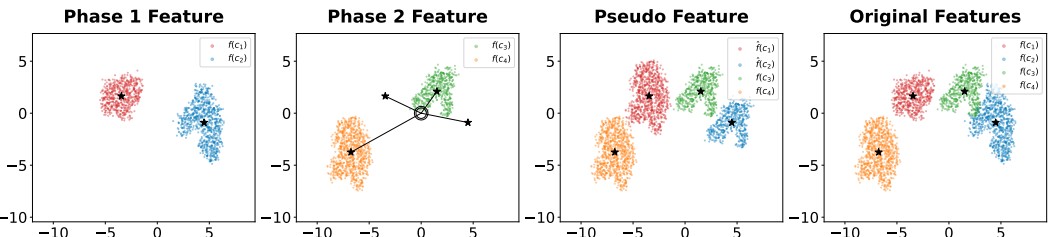

Figure 3: Illustration of the pseudo-feature generation procedure with a toy example. In Phase 1, the model distinguishes between red and blue classes using actual features. In Phase 2, it learns to discriminate among all seen classes (red, blue, orange, green) without access to past-class data. For each past class, the closest new class is identified by cosine similarity between centroids (marked by the ⋆ symbol). Pseudo-features for past classes are generated by shifting the feature distribution from the closest new class to the target past class. The model then distinguishes all seen classes using actual features for new classes and pseudo-features for past classes. The rightmost subfigure visualizes the actual features for all classes at the end of Phase 2.

**Module (III): Dynamic Adaptation of Prediction Layer and Pseudo-Concept Generation.** The prediction layer, $W_F^t$, and its bias, $b_F^t$, must be expanded to accommodate both new concepts and class labels, while preserving the learned associations between previous classes and concepts stored in $W_F^{t-1}$ and $b_F^{t-1}$. Following Petit et al. (2023), we generate pseudo-features for past classes by shifting the data distribution of the nearest new class from the new class mean to the target past class mean, as shown in Figure 3. The centroids of each class in the backbone's feature space are computed and stored as the class is introduced. For each past class, its nearest new class is identified by calculating the cosine similarity between their centroids. Let $c_p$ denote each past class, $c_n$ the closest new class to $c_p$, and $\mu(c_p)$ and $\mu(c_n)$ the mean features of classes $c_p$ and $c_n$, respectively, as extracted by the frozen backbone model $f$. Pseudo-features for past classes are generated by the following shift:

$$\hat{f}(c_p) = f(c_n) - \mu(c_n) + \mu(c_p) \tag{4}$$

The pseudo-features are then projected into the concept space using the learned concept bottleneck at phase $t$, $W_C^t$, to generate pseudo-concepts:

$$\hat{f}_c(c_p) = W_C^t \hat{f}(c_p) \tag{5}$$

These pseudo-concepts for past classes, along with the actual concepts for new classes, are used to train the sparse predictor $W_F^t$:

$$\min_{W_F^t, b_F^t} \sum_{(x_i,y_i)\in D_1\cup\cdots\cup D_{t-1}} L_{ce}(W_F^t \hat{f}_c(x_i) + b_F^t, y_i) + \sum_{(x_i,y_i)\in D_t} L_{ce}(W_F^t f_c(x_i) + b_F^t, y_i) + \lambda R_\alpha(W_F^t) \tag{6}$$

This allows us to train the model with pseudo-concepts for previous classes, without storing any information except for each class mean. See Algorithm 1 for the full sequence of steps across phases.

### 3.3 Controlled Geometric Perspective

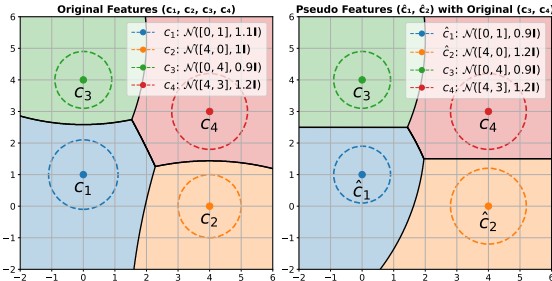

Figure 4: Visualization of decision boundaries in the feature space. Black lines indicate Bayes-optimal boundaries, and colored regions denote predicted class assignments. Dashed circles represent one standard deviation from each class mean. Left: original class feature distributions, where differences in variance lead to curved boundaries between old and new classes. Right: pseudo-feature distributions generated by our method, where matched variances lead to linear boundaries that closely follow the true Bayes boundaries.

To further motivate the design of our method, we provide a controlled analysis based on class distributions in the feature space. Consider a toy example with two learning phases, each introducing two new classes. We model each class's feature distribution as a multivariate Gaussian $\mathcal{N}(\mu_i, \sigma_i^2 I)$ where $x \in \mathbb{R}^d$. The Bayes optimal decision boundary between classes $i$ and $j$ is where $\log p_i(x) = \log p_j(x)$:

$$\log p_k(x) = -\frac{d}{2}\log(2\pi\sigma_k^2) \; - \; \frac{1}{2\sigma_k^2}\|x-\mu_k\|^2,$$

$$\log p_i(x) = \log p_j(x) \iff \frac{1}{\sigma_i^2}\|x-\mu_i\|^2 + d\log\sigma_i^2 = \frac{1}{\sigma_j^2}\|x-\mu_j\|^2 + d\log\sigma_j^2$$

$$\iff \left(\frac{1}{\sigma_j^2}-\frac{1}{\sigma_i^2}\right)x^\top x - 2\left(\frac{\mu_j}{\sigma_j^2}-\frac{\mu_i}{\sigma_i^2}\right)^\top x + \left(\frac{\|\mu_j\|^2}{\sigma_j^2}-\frac{\|\mu_i\|^2}{\sigma_i^2}\right) = d\log\left(\frac{\sigma_j^2}{\sigma_i^2}\right).$$

This boundary is generally quadratic but becomes approximately linear when class variances are similar. Our method shifts new class distributions toward old class centroids, forming pseudo-feature distributions with original means and nearby new class variances—e.g., class 1 becomes $\mathcal{N}(\mu_1, \sigma_3^2 I)$, enabling a linear boundary with class 3 due to matched variance. The boundary between real and pseudo-feature distributions closely matches the true boundary, especially when their variances are similar. Figure 4 illustrates this effect. The left panel shows the Bayes-optimal decision regions induced by the original class distributions, where differences in class variances lead to curved boundaries. In contrast, the right panel depicts the pseudo-feature distributions introduced by our method, where the resulting boundaries become linear while still closely approximating the true Bayes decision surfaces. Subsequently, all features (actual features for new classes and pseudo-features for old classes) are projected into the concept space, where concept functionality is preserved via our drift mitigation, enabling concept-based classification.

This controlled analysis assumes that class-conditional features are reasonably concentrated around their centroids, so that mean shifts preserve much of the relevant geometry. This requires a well-trained feature extractor, which may not always be available in realistic CIL settings. To bridge this controlled picture to realistic high-dimensional representations and to assess sensitivity to pretrained vs. non-pretrained feature extractors, we provide an empirical validation in Supplementary Section A15.

## 4 Evaluation

**Datasets and Implementation Details.** To evaluate the performance of our proposed method, we perform comprehensive experiments on several CIL datasets: CIFAR-10, CIFAR-100 (Krizhevsky et al., 2009), CUB (Wah et al., 2011), TinyImageNet (Le & Yang, 2015), ImageNet-Subset, ImageNet (Deng et al., 2009), and Places365 (Zhou et al., 2017). We employ ResNet-18 (He et al., 2016), DeiT (Touvron et al., 2021), modified to match ResNet-18's parameter count by adjusting the embedding dimension and number

| Method | CIFAR-10 | CIFAR-100 | | | CUB | | | TinyImageNet | | | Places365 | | | ImageNet | | |
|---|---|---|---|---|---|---|---|---|---|---|---|---|---|---|---|---|
| | $T=5$ | $T=5$ | $T=10$ | $T=20$ | $T=4$ | $T=10$ | $T=20$ | $T=5$ | $T=10$ | $T=20$ | $T=5$ | $T=10$ | $T=20$ | $T=5$ | $T=10$ | $T=20$ |
| ICICLE (Rymarczyk et al., 2023) | - | - | - | - | 35.0 | 18.5 | 9.9 | - | - | - | - | - | - | - | - | - |
| IN2 (Yang et al., 2024) | 44.9 | 41.9 | 27.5 | 17.3 | 30.5 | 20.1 | 13.6 | 30.9 | 21.4 | 14.3 | - | - | - | - | - | - |
| CONCIL (Lai et al., 2025) | - | - | - | - | - | 61.3 | - | - | - | - | - | - | - | - | - | - |
| CI-CBM (ours) | **80.4** | **68.8** | **68.8** | **67.8** | **62.2** | **65.3** | **66.1** | **48.6** | **48.7** | **48.5** | **48.3** | **49.3** | **49.0** | **31.7** | **32.1** | **31.5** |
| Full rehearsal | 85.9 | 70.9 | 72.6 | 73.4 | 67.7 | 67.2 | 67.1 | 50.0 | 51.4 | 52.5 | 48.9 | 50.7 | 51.6 | 32.1 | 33.3 | 34.1 |

Table 1: **Experiment I -** Comparisons of the Average Incremental Accuracy of CI-CBM with other interpretable models in CIL. The same pretrained backbone was used to reproduce IN2 (Yang et al., 2024) results for a fair comparison. The results for ICICLE (Rymarczyk et al., 2023) and CONCIL (Lai et al., 2025) are as reported in their original paper. Cells marked with "-" indicate that results were unavailable. The full rehearsal approach retains all previous training data across phases, providing an upper bound for evaluating CI-CBM's performance.

of heads, and ViT-Base/16 (Dosovitskiy et al., 2020) as backbone models. The subsequent parameters follow the setup from LF-CBM (Oikarinen et al., 2023). The regularization parameter $\lambda$ is selected to ensure that each class has 35 to 55 non-zero weights, leading to sparsity levels ranging from 1% to 5% across different datasets. To ensure a fair comparison, we use the same random seed `1993` as the compared methods (Zhu et al., 2021b; 2022; Petit et al., 2023; Rymarczyk et al., 2023) to shuffle the classes and split them into phases. Each configuration is evaluated three times, and the average results are reported.

**Evaluation Metric.** We report the average incremental accuracy as the mean of the average accuracy across all incremental phases, including the first phase. Additionally, we offer average accuracy for each phase to provide an understanding of the accuracy progression throughout the continual learning process. Further, we report the average incremental forgetting in the Appendix.

**Comparison to Interpretable Methods.** We compare our interpretable model against ICICLE, IN2, CONCIL, and CLG-CBM [1], which, to the best of our knowledge, are the only existing interpretable methods designed for the CIL setting. We also evaluate a full rehearsal strategy, where the training data from all previous phases is stored in memory and accessible for each new phase, though this is highly impractical in real-world settings. However, this approach helps us understand how closely our method reaches the best possible performance. Our backbone model for the CIFAR-10/100, CUB, and Places365 datasets is ResNet-18 pretrained on ImageNet, and for the TinyImageNet and ImageNet datasets, it is pretrained on Places365. For each dataset, we distribute the classes evenly across $T$ phases.

Table 1 (Experiment I) demonstrates that our approach significantly outperforms all the compared interpretable methods. CI-CBM achieves an average improvement of 35.50%, 39.57%, and 26.40% on CIFAR-10, CIFAR-100, and TinyImageNet, respectively. On CUB, CI-CBM improves over prior interpretable baselines (ICICLE[2]/IN2) by 43.13% on average across $T \in \{4, 10, 20\}$[3], and additionally outperforms CONCIL at their reported setting ($T = 10$) by 4.0%. Our performance on the Places365 and ImageNet datasets demonstrates that our proposed approach is capable of incremental classification on challenging large-scale datasets. As the total number of phases increases, our method remains robust and maintains its average incremental accuracy, while the other compared methods fail to do so. By comparing CI-CBM with the impractical full rehearsal scenario, we find that our approach shows only a small 2.6% decrease on average, demonstrating its effectiveness in maintaining learned knowledge without retaining any samples from previous phases.

**Comparison to Non-Pretrained and Non-Interpretable Methods**. Motivated by the results observed in interpretable models and the investigation of the effect of the pretrained backbone model on our metric scores, we extend our analysis to unrestricted models. We follow FeTrIL (Petit et al., 2023) and APG (Tang

---

[1]CLG-CBM uses a pretrained ViT backbone; therefore, we report the comparison to CLG-CBM under the same pretrained ViT setting in Table 4. Table 1 focuses on comparisons under a ResNet backbone for fair evaluation among ResNet-based interpretable methods.

[2]ICICLE used a ResNet-34 backbone pre-trained on ImageNet. We use ResNet-18 for compatibility with most CIL models. With ResNet-34, our results are 62.9, 65.9, and 66.4 for T = 4, 10, and 20 phases, respectively.

[3]We note that on CUB the average incremental accuracy slightly increases as the number of phases grows. This can occur because an ImageNet-pretrained backbone yields a well-structured feature space, allowing pseudo-features to preserve past knowledge effectively. Since all classes in CUB are birds, the concepts introduced in each phase remain broadly relevant and discriminative, which can further improve average incremental performance as phases become finer-grained.

| Method | Interpretability | CIFAR-100 | | | TinyImageNet | | | ImageNet-Subset | | |
|---|---|---|---|---|---|---|---|---|---|---|
| | | $T=5$ | $T=10$ | $T=20$ | $T=5$ | $T=10$ | $T=20$ | $T=5$ | $T=10$ | $T=20$ |
| EWC[†] (Kirkpatrick et al., 2017) | ✗ | 24.5 | 21.2 | 15.9 | 18.8 | 15.8 | 12.4 | - | 20.4 | - |
| LWF[§] (Li & Hoiem, 2017) | ✗ | 32.4 | 17.9 | 14.9 | 22.3 | 17.4 | 12.5 | - | 23.5 | - |
| iCaRL-CNN[§] (Rebuffi et al., 2017) | ✗ | 51.0 | 48.3 | 44.6 | 34.7 | 31.0 | 27.8 | - | 50.5 | - |
| LUCIR[†] (Hou et al., 2019) | ✗ | 51.2 | 41.1 | 25.2 | 41.7 | 28.1 | 18.9 | 56.8 | 41.4 | 28.5 |
| MUC[†] (Liu et al., 2020) | ✗ | 49.4 | 30.2 | 21.3 | 32.6 | 26.6 | 21.9 | - | 35.1 | - |
| SDC[†] (Yu et al., 2020) | ✗ | 56.8 | 57.0 | 58.9 | - | - | - | - | 61.2 | - |
| PASS[†] (Zhu et al., 2021b) | ✗ | 63.5 | 61.8 | 58.1 | 49.6 | 47.3 | 42.1 | 64.4 | 61.8 | 51.3 |
| SSRE[†] (Zhu et al., 2022) | ✗ | 65.9 | 65.0 | 61.7 | 50.4 | 48.9 | 48.2 | - | 67.7 | - |
| FeTrIL[†] (Petit et al., 2023) | ✗ | 66.3 | 65.2 | 61.5 | 54.8 | 53.1 | 52.2 | 72.2 | 71.2 | 67.1 |
| EFC * (Magistri et al., 2024) | ✗ | - | 68.2 | 65.9 | - | 57.5 | 56.5 | - | 75.4 | 71.6 |
| SOPE* (Zhu et al., 2023) | ✗ | 66.6 | 65.8 | 61.8 | 53.7 | 52.9 | 51.9 | - | 69.2 | - |
| FCS* (Li et al., 2024) | ✗ | 62.1 | 60.3 | 58.3 | 46.0 | 44.9 | 42.5 | - | 61.7 | - |
| DCMI* (Qiu et al., 2024) | ✗ | 67.9 | 66.8 | 64.0 | 54.8 | 53.9 | 52.5 | 70.5 | 70.0 | 65.5 |
| TASS* (Liu et al., 2024) | ✗ | 68.7 | 67.4 | 62.7 | 55.1 | 54.2 | 52.7 | 74.3 | 72.6 | 68.7 |
| **CI-CBM (ours)** | ✓ | 61.9 | 60.5 | 60.2 | 49.2 | 48.3 | 47.3 | 67.6 | 66.2 | 66.9 |
| Full rehearsal | ✓ | 63.7 | 63.5 | 62.1 | 50.5 | 50.4 | 50.2 | 71.7 | 71.7 | 69.6 |

Table 2: **Experiment II -** Comparisons of the average incremental accuracy of CI-CBM with EFCIL unrestricted ResNet-based methods in a non-pretrained scenario. Models marked with a † represent reported results from (Petit et al., 2023), while those marked with an § indicate reported findings from (Zhu et al., 2021b), and models marked with a ∗ denote results reported in their paper. Cells marked with "-" indicate that results were unavailable. The full rehearsal approach retains all previous training data across phases, providing an upper bound for evaluating CI-CBM's performance.

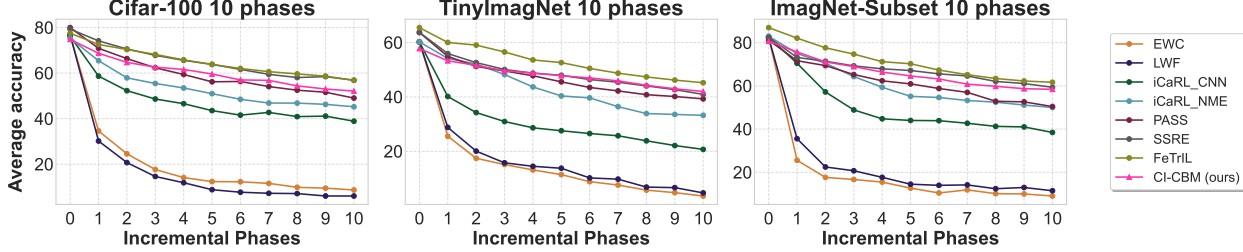

Figure 5: **Experiment III -** Average accuracy curves for CIFAR-100, TinyImageNet, and ImageNet-Subset over 10 learning phases, comparing CI-CBM with other unrestricted ResNet-based methods.

et al., 2023) to train ResNet-18 and DeiT, respectively, from scratch in the initial phase. Afterward, we freeze them as the backbone for CI-CBM and incrementally learn the classes. We conduct a comparative analysis of our method against classical approaches (Kirkpatrick et al., 2017; Li & Hoiem, 2017; Rebuffi et al., 2017; Hou et al., 2019), SOTA ResNet-based models (Zhu et al., 2021b;a; 2022; Petit et al., 2023), and SOTA prompt-based methods (Wang et al., 2022c;b; Tang et al., 2023).

We follow the setup described in (Petit et al., 2023; Tang et al., 2023), evaluating CIFAR-100 and ImageNet-Subset under the following configurations: (i) an initial set of 50 classes with 5 phases, each introducing 10 classes, (ii) 50 initial classes followed by 10 phases of 5 classes each, (iii) 40 initial classes with 20 phases, each introducing 3 classes. Additionally, TinyImageNet is tested with 100 initial classes, with the remaining classes distributed as follows: (i) 5 phases of 20 classes, (ii) 10 phases of 10 classes, (iii) 20 phases of 5 classes.

Table 2 (Experiment II) illustrates that our interpretable model demonstrates superior performance compared to many unrestricted ResNet-based models, with a minimal 7.5% difference in accuracy compared to the SOTA models. Figure 5 (Experiment III) presents the average accuracy curve for CIFAR-100, TinyImageNet, and ImageNet-Subset across 10 phases. Although CI-CBM starts with the lowest accuracy in the initial phase due to interpretability constraints, it effectively learns to distinguish new classes while preserving performance on previously learned classes, ultimately achieving a high average accuracy by the final phase.

Table 3 (Experiment IV) compares the performance of CI-CBM with prompt-based models using a DeiT backbone trained from scratch in the first phase. Without needing any additional prompts, CI-CBM outper-

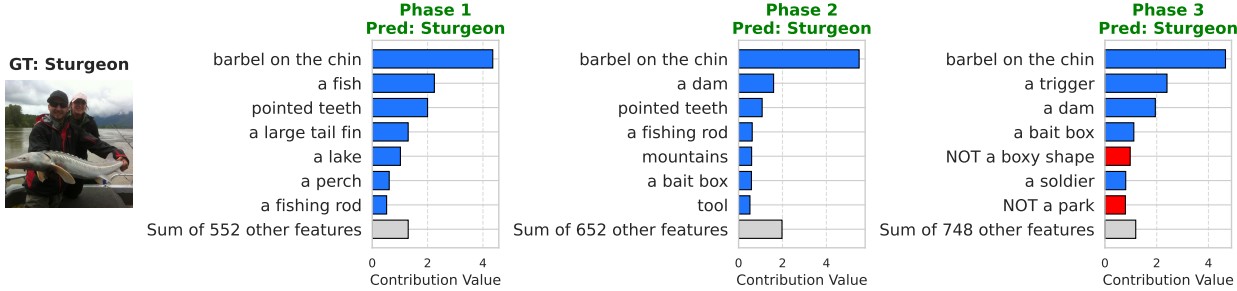

Figure 6: Visualization of model reasoning and concept contributions for an image of the Sturgeon class, introduced in the first phase of the ImageNet-Subset. In the first phase, CI-CBM correctly classifies the image using actual features. In the second and third phases, CI-CBM continues to accurately distinguish it among a larger set of classes by generating pseudo-concepts. Additional visualizations are provided in Supplement Section A16.

forms L2P (Wang et al., 2022c) and DualPrompt (Wang et al., 2022b), two methods that fail to generalize when trained from scratch in the initial phase. CI-CBM achieves performance comparable to APG (Tang et al., 2023), despite the latter's use of prompt generators, which involve cross-attention layers, groups of learnable parameters, and linear layers, leading to a more parameter-intensive model. Results with more incremental phases are reported in the Appendix.

**Comparison to Pretrained ViT-Based Methods.** Following prior works, we use a ViT-Base/16 model pretrained on ImageNet-21k, utilizing the final [CLS] token as the backbone feature. For each dataset, we distribute the classes evenly across $T$ phases. Table 4 (Experiment V) shows that CI-CBM achieves competitive accuracy, only 7% lower on CIFAR-100 and 3.1% lower on CUB compared to SOTA models, demonstrating that CI-CBM provides interpretability with minimal performance trade-off. Moreover, CI-CBM outperforms the interpretable baseline CLG-CBM by 1.1% on CIFAR-100 and 2.6% on CUB.

**Interpretability and Insights on Model Reasoning.** To provide a comprehensive understanding of our model's interpretability, Figure 1 presents a Sankey diagram that offers global insights into the concept-to-class relationships. In the diagram, line widths are proportional to absolute weights, displaying only concepts with absolute weights greater than 0.2. Concepts with negative weights are labeled as "NOT" concepts. The findings reveal that the model consistently relied on the same positive concepts across different phases without needing to impose constraints like freezing weights. Additionally, as new classes were introduced, more discriminative negative concepts were incorporated, enabling the model to better differentiate between similar classes. Figure 6 illustrates the local explanations for individual decisions made by the model using a randomly chosen image from the Sturgeon class in the first phase of the ImageNet-Subset dataset. The contribution of concept $j$ to the output $i$ for input $x_k$ in phase $t$ is given by $\text{Contrib}(x_k, i, j) = W_F^t[i, j] \times f_c^t(x_k)[j]$. This figure illustrates how the model's salient concepts evolve across phases: as semantically related classes (e.g., eel, coho) are introduced, generic concepts (e.g., "a fish") become less discriminative and diminish in contribution, while more class-specific concepts (e.g., "barbel on the chin") remain dominant, allowing the model to classify the image correctly in later phases.

| Method | Interpretability | CIFAR-100 | | ImageNet-Subset | |
|---|---|---|---|---|---|
| | | $T=10$ | $T=20$ | $T=10$ | $T=14$ |
| L2P (Wang et al., 2022c) | ✗ | 36.5 | 18.8 | 25.1 | 29.9 |
| DualPrompt (Wang et al., 2022b) | ✗ | 26.8 | 11.8 | 35.8 | 30.3 |
| APG (Tang et al., 2023) | ✗ | 66.6 | 62.4 | 75.5 | 69.8 |
| **CI-CBM (ours)** | ✓ | 59.5 | 59.7 | 54.6 | 55.6 |
| Full rehearsal | ✓ | 62.7 | 63.4 | 58.9 | 60.4 |

Table 3: **Experiment IV –** Comparison of the average incremental accuracy between CI-CBM and EFCIL unrestricted prompt-based methods in a non-pretrained scenario. The DeiT backbone model is trained using data from the first phase. Results for the compared methods are reported from (Tang et al., 2023).

| Method | Interpretability | CIFAR-100 ($T$=10) | CUB ($T$=10) |
|---|:---:|:---:|:---:|
| L2P[†] (Wang et al., 2022c) | ✗ | 84.6 | 65.2 |
| DualPrompt[†] (Wang et al., 2022b) | ✗ | 81.3 | 68.5 |
| CODA-Prompt[†] (Smith et al., 2023) | ✗ | 86.3 | 79.5 |
| APG * (Tang et al., 2023) | ✗ | 89.3 | - |
| LAE *(Gao et al., 2023) | ✗ | 89.9 | - |
| ESN *(Wang et al., 2023) | ✗ | 86.3 | - |
| SimpleCIL[†] (Zhou et al., 2024a) | ✗ | 87.6 | 87.1 |
| ConvPrompt * (Roy et al., 2024) | ✗ | 88.8 | 80.2 |
| EASE[†] (Zhou et al., 2024c) | ✗ | 87.8 | 86.8 |
| SAFE[†] (Zhao et al., 2024) | ✗ | 92.8 | 91.1 |
| CLG-CBM (Yu et al., 2025)* | ✓ | 84.5 | 85.4 |
| **CI-CBM (ours)** | ✓ | 85.6 | 88.0 |
| Full rehearsal | ✓ | 87.8 | 88.7 |

Table 4: **Experiment V -** Average incremental accuracy of CI-CBM and EFCIL with a pretrained ViT-Base/16 backbone. † indicates results reported by Zhao et al. (2024); ∗ indicates results taken from the corresponding papers; "−" denotes unavailable results.

| Concept reg | Pseudo-Concept | CIFAR-100 | | | TinyImageNet | | |
|:---:|:---:|:---:|:---:|:---:|:---:|:---:|:---:|
| | | $T$=5 | $T$=10 | $T$=20 | $T$=5 | $T$=10 | $T$=20 |
| ✗ | ✗ | 38.0 | 26.2 | 16.9 | 29.1 | 20.9 | 14.4 |
| ✓ | ✗ | 38.0 | 26.3 | 16.9 | 29.2 | 21.0 | 14.3 |
| ✗ | ✓ | 68.4 | 68.2 | 67.6 | 48.2 | 48.3 | 48.2 |
| ✓ | ✓ | **68.8** | **68.8** | **67.8** | **48.6** | **48.7** | **48.5** |

Table 5: **Experiment VI (Ablation Study) -** Performance impact of different components of CI-CBM

**Ablation Study.** To assess the effect of each component in CI-CBM, we conduct an ablation study on CIFAR-100 and TinyImageNet datasets, analyzing the average incremental accuracy. Table 5 (Experiment VI) highlights the contributions of concept regularization and pseudo-concept generation. The results are as follows: (i) Skipping pseudo-concept generation leads to weights with zero and negative biases in previous class weights, causing their accuracy to drop to zero and significantly lowering overall performance. (ii) Concept regularization alone cannot prevent this, as it only preserves concepts within the Concept Bottleneck Layer but does not stop the weights for previous classes from being pushed to zero. (iii) Pseudo-concepts improve performance by helping the model distinguish between new and old classes. (iv) Combining pseudo-concept generation with concept regularization provides further performance gains. In the Appendix, we include a broader set of ablations covering concept generation and filtering choices, predictor sparsity and concept-set size, robustness to noisy concept activations, and an alternative strategy for learning the prediction layer. We further study sensitivity to the distillation weight $\beta$ (both accuracy and concept-fidelity metrics), the effect of swapping CLIP with SigLIP when computing the concept-activation matrix $P$ (accuracy and fidelity), the unique-concept expansion mechanism and its impact on model size/interpretability, and pseudo-feature reliability under ImageNet-pretrained vs. phase-1-trained backbones.

# 5    Conclusion

In this work, we propose **CI-CBM**, an interpretable model for Exemplar-Free Class Incremental Learning (EFCIL). **CI-CBM** extends the Concept Bottleneck Model with concept regularization and pseudo-concept generation to incrementally learn the concept bottleneck layer and sparse prediction layer. Our approach outperforms other interpretable models designed for EFCIL, achieving an average accuracy gain of 36% and approaching SOTA black-box performance, demonstrating effectiveness in both pretrained and non-pretrained settings. We further illustrate how **CI-CBM**'s decision-making adapts with new phases, providing global and local insights, and present ablation studies to assess the impact of various model components.

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

## A1 Appendix Overview

In this section, we provide a brief overview of the Appendix contents. The Appendix is primarily focused on (i) defining precise evaluation metrics for the CIL setting, (ii) presenting extended experiments and ablation studies that analyze the behavior of CI-CBM across different backbones and regimes, and (iii) showcasing additional qualitative visualizations that illustrate model reasoning and weight structure.

First, in Section A2, we discuss the main limitations of our method. Section A3 defines the metrics used throughout our CIL evaluations and reports extended results on average incremental forgetting across three backbones and datasets (Experiments VII–IX). Sections A4–A8 present a series of ablation studies examining concept generation, sparsity, concept-set size, robustness to noisy concept activations, and an alternative strategy for learning the prediction layer (Experiments X–XIV). In Section A9, we analyze our unique-concept expansion mechanism and its effect on model size and interpretability (Experiment XVI). Section A10 evaluates the challenging one-class-increment regime (Experiment XV). Section A11 details the full concept-generation pipeline, including prompts, few-shot examples, and filtering steps. Section A12 studies the effect of the distillation regularizer weight on class-incremental accuracy (Experiment XVII; Table A10), while Section A13 analyzes how distillation impacts the interpretability and semantic fidelity of the learned concept bottleneck using CLIP-based concept-fidelity metrics. We further ablate the choice of vision-language model used to compute the concept-activation matrix $P$ by swapping CLIP with SigLIP, and report its effect on both incremental accuracy (Experiment XIX; Table A12) and concept fidelity (Experiment XX; Table A13). Section A15 analyzes pseudo-feature reliability using a cosine-prototype classifier under two backbone regimes (ImageNet-pretrained vs. trained only on the first phase), highlighting the effect of pretraining on feature separability and accuracy, and showing that our full method improves test accuracy in both regimes. Finally, Section A16 provides additional qualitative visualizations of model weights and per-image concept attributions that complement the figures in the main paper.

## A2 Limitation

The main limitations of our research point to future directions. First, the model sometimes forms incorrect correlations between specific concepts and classes, warranting further investigation to see if this stems from the training process or vision-language model misalignments. Second, our pseudo-concept generation relies on basic geometric translations and projections. A more refined approach to generating realistic pseudo-concepts could reduce overlap between old and new classes, improving model performance.

## A3 Metrics for the CIL Setting and Further Experiments

Let $a_{i,j}$ denote the model's accuracy on the $j$-th task after learning the $i$-th task, where $i \geqslant j$. The following standard metrics are used to evaluate continual learning performance, assuming each phase contains an equal amount of data:

- Average Phase Accuracy:

$$A_t = \frac{1}{t} \sum_{j=1}^{t} a_{t,j} \tag{A1}$$

- Average Phase Forgetting:

$$F_t = \frac{1}{t-1} \sum_{j=1}^{t-1} \max_{i \in \{1,\ldots,t-1\}} (a_{i,j} - a_{t,j}) \tag{A2}$$

- Average Incremental Accuracy:

$$\bar{A} = \frac{1}{T} \sum_{t=1}^{T} A_t \tag{A3}$$

- Average Incremental Forgetting:

$$\bar{F} = \frac{1}{T-1} \sum_{t=2}^{T} F_t \tag{A4}$$

For scenarios with unbalanced phases, such as when the first phase contains a larger number of classes for backbone pretraining, a weighted version of average phase accuracy and forgetting should be calculated, where the weights are proportional to the size of each phase.

| Method | CIFAR-100 | | | | TinyImageNet | | | | ImageNet-Subset | | | |
|---|---|---|---|---|---|---|---|---|---|---|---|---|
| | $T$=5 | $T$=10 | $T$=20 | $T$=60 | $T$=5 | $T$=10 | $T$=20 | $T$=100 | $T$=5 | $T$=10 | $T$=20 | $T$=60 |
| CI-CBM | 11.4 | 13.0 | 17.1 | 19.3 | 8.4 | 9.5 | 10.9 | 14.6 | 11.9 | 13.6 | 16.9 | 19.5 |
| Full rehearsal | 8.8 | 8.7 | 11.0 | 11.7 | 7.0 | 7.2 | 7.4 | 7.8 | 7.0 | 7.1 | 8.1 | 8.6 |

Table A1: **Experiment VII -** Average incremental forgetting of CI-CBM in the non-pretrained scenario with the ResNet-18 backbone, trained on first-phase data from CIFAR-100, TinyImageNet, and ImageNet-Subset, and evaluated under 5, 10, 20, and 60 incremental phase settings. The full rehearsal approach retains all previous training data across phases, providing an upper bound for evaluating CI-CBM's performance. (Complementary to the results in Table 2).

| Method | $T$ | CIFAR-100 | | ImageNet-Subset | |
|---|---|---|---|---|---|
| | | $\bar{A}$ | $\bar{F}$ | $\bar{A}$ | $\bar{F}$ |
| CI-CBM (ours) | 5 | 60.1 | 8.5 | 56.2 | 6.6 |
| | 10 | 59.5 | 8.5 | 54.6 | 8.9 |
| | 20 | 59.7 | 10.2 | 54.4 | 10.8 |
| | 60 | 58.5 | 11.7 | 52.3 | 13.4 |
| Full rehearsal | 5 | 62.5 | 4.4 | 58.6 | 4.1 |
| | 10 | 62.7 | 4.5 | 58.9 | 4.6 |
| | 20 | 63.4 | 5.1 | 59.6 | 5.5 |
| | 60 | 63.4 | 6.0 | 59.0 | 6.4 |

Table A2: **Experiment VIII -** Average incremental accuracy ($\bar{A}$) and forgetting ($\bar{F}$) of CI-CBM in the non-pretrained scenario with the DeiT backbone, trained on first-phase data from CIFAR-100 and ImageNet-Subset, and evaluated under 5, 10, 20, and 60 incremental phase settings. The full rehearsal approach retains all previous training data across phases, providing an upper bound for evaluating CI-CBM's performance. (Complementary to Table 3).

| Method | $T$ | CIFAR-100 | | CUB | |
|---|---|---|---|---|---|
| | | $\bar{A}$ | $\bar{F}$ | $\bar{A}$ | $\bar{F}$ |
| CI-CBM (ours) | 5 | 85.6 | 6.5 | 87.6 | 4.9 |
| | 10 | 85.6 | 7.9 | 88.0 | 4.5 |
| | 20 | 85.3 | 9.0 | 88.2 | 4.6 |
| Full rehearsal | 5 | 87.0 | 3.3 | 88.5 | 4.0 |
| | 10 | 87.8 | 3.4 | 88.7 | 3.5 |
| | 20 | 88.2 | 3.8 | 88.7 | 4.2 |

Table A3: **Experiment IX -** Average incremental accuracy ($\bar{A}$) and forgetting ($\bar{F}$) of CI-CBM in the pretrained scenario with the ImageNet-pretrained ViT-B/16 backbone, evaluated under 5, 10, and 20 incremental phase settings. The full rehearsal approach retains all previous training data across phases, providing an upper bound for evaluating CI-CBM's performance. (Complementary to Table 4).

Tables A1 (Experiment VII) and A2 (Experiment VIII) present the average incremental forgetting in the non-pretrained scenario with the ResNet-18 and DeiT backbones, respectively. Table A3 (Experiment IX) presents the average incremental forgetting in the pretrained scenario with the ViT-B/16-IN21K backbone.

## A4 Alternative Concept Generation Methods

We conduct an experiment to evaluate the importance of using GPT-3 for our model's performance by comparing it against generating the initial concept sets with ConceptNet (Speer & Havasi, 2013). Note that ConceptNet is not a language model but rather a knowledge graph. The results in Table A4 (Experiment X) show that our proposed pipeline, when using the concept sets generated from ConceptNet, still performs well on CIFAR-100 and TinyImageNet. However, there is a slight decrease in accuracy (approximately a 2% drop) compared to using GPT-3-generated concepts. Furthermore, we observe that ConceptNet completely

fails on CUB, whereas GPT-3-generated concepts achieve strong results. This highlights the effectiveness of GPT-3 in generating concepts for fine-grained datasets where ConceptNet struggles.

| Method | CIFAR-100 | | | CUB | | | TinyImageNet | | |
|---|---|---|---|---|---|---|---|---|---|
| | $T{=}5$ | $T{=}10$ | $T{=}20$ | $T{=}5$ | $T{=}10$ | $T{=}20$ | $T{=}5$ | $T{=}10$ | $T{=}20$ |
| CI-CBM (ConceptNet) | 66.8 | 67.1 | 66.5 | 27.6 | 24.3 | 23.8 | 48.3 | 48.4 | 47.8 |
| CI-CBM (GPT-3 [original]) | **68.8** | **68.8** | **67.8** | **62.2** | **65.3** | **66.1** | **48.6** | **48.7** | **48.5** |

Table A4: **Experiment X (Ablation Study) -** ConceptNet vs. GPT-3 for initial concept set generation

## A5   Impact of Sparsity on Model Performance

Table A5 (Experiment XI) presents the effect of sparsity along with the corresponding sparsity levels. Wong et al. (2021) proposed fitting a sparse linear prediction layer on top of deep feature representations, showing that sparse models are more interpretable while maintaining high accuracy. Our results generally suggest that removing the sparsity constraint neither improves performance nor interpretability. In particular, CI-CBM with a dense prediction layer (i.e., setting $\lambda = 0$ in Eq. 2) performs slightly worse than the sparse variant across both CIFAR-100 and TinyImageNet (Table A5). During training, our goal is to optimize the prediction layer to distinguish between pseudo-concepts of past classes and actual concepts of new classes, expecting that the learned layer will classify both old and new classes based on their actual concepts in the test datasets. A dense layer might overly focus on the pseudo-concept distribution, while a sparse prediction layer relies on fewer concepts per class, making it more robust and yielding slightly better performance.

| Method | CIFAR-100 | | | TinyImageNet | | |
|---|---|---|---|---|---|---|
| | $T{=}5$ | $T{=}10$ | $T{=}20$ | $T{=}5$ | $T{=}10$ | $T{=}20$ |
| CI-CBM (dense) | 68.4 | 68.6 | **68.3** | 46.5 | 47.0 | 47.0 |
| CI-CBM (sparse [original]) | **68.8** | **68.8** | 67.8 | **48.6** | **48.7** | **48.5** |
| Sparsity | 4.84% | 5.70% | 7.02% | 2.06% | 2.43% | 3.36% |

Table A5: **Experiment XI (Ablation Study) -** Performance impact of sparsity constraints

Beyond accuracy, we also evaluate interpretability when removing sparsity. Figure A1 repeats the same per-image concept contribution analysis used in Fig. 6, but trains the final predictor without the sparsity regularizer. Although the dense model still classifies this example correctly, the attribution mass is spread across many concepts: the top-ranked concepts each contribute only marginally to the predicted class, while the aggregate contribution of the remaining concepts dominates. This makes the explanation less concise and harder to interpret, since there is no small set of core concepts with clearly dominant contributions.

## A6   Effect of Concept Set Size on Performance

Table A6 (Experiment XII) presents the impact of reducing the number of available concepts during training. We simulate reduced concept availability by randomly masking a portion of the concept set and training the model using only the remaining subset. This setting evaluates the model's robustness to incomplete or noisy concept supervision. As shown, performance remains relatively stable even when only 25% of the concept set is used, with less than a 3.5% drop in accuracy for CIFAR-100 and a 2.1% drop for TinyImageNet. Beyond predictive performance, we investigate whether reducing the available concept set degrades explanation quality. We repeat the qualitative interpretability analyses under a reduced-concept setting in which 50% of the concepts are randomly masked. Fig. A2 presents the final-layer weight structure for the Tree Swallow class on CUB across four phases under 50% concept availability, corresponding to the full-concept visualization in Fig. 1. Although some class-specific concepts (e.g., blue head and small, forked tail) are unavailable under masking, the model selects semantically and visually related alternatives (e.g., blue eyes and

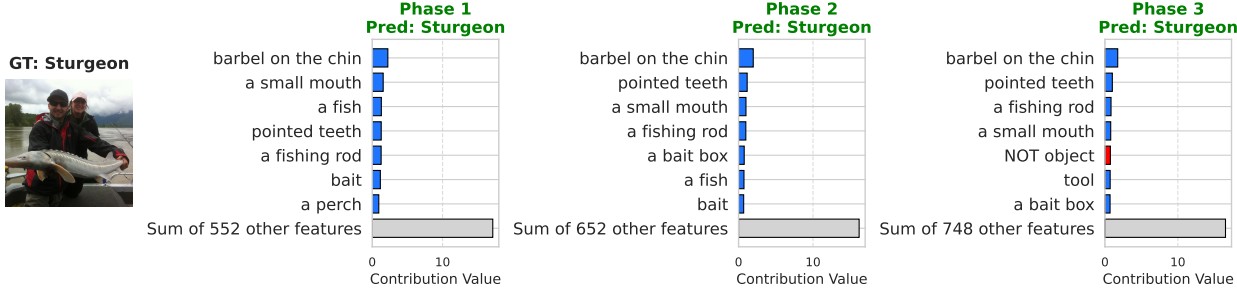

Figure A1: Visualization of model reasoning and concept contributions for the same Sturgeon example as Fig. 6, but with a dense prediction layer trained by setting the sparsity coefficient to $\lambda = 0$ in Eq. 2. Although the prediction is correct, the contribution is dispersed across many concepts: the top-ranked concepts have small individual impact compared to the summed contribution of the remaining concepts, making the explanation less concise and harder to interpret.

|  | CIFAR-100 | | | | TinyImageNet | | | |
|---|---|---|---|---|---|---|---|---|
| Concept Use | 100% | 75% | 50% | 25% | 100% | 75% | 50% | 25% |
| Accuracy | 68.8 | 68.4 | 67.5 | 65.5 | 48.6 | 48.0 | 47.5 | 46.5 |

Table A6: **Experiment XII (Ablation Study) -** Accuracy vs. concept availability.

small, rounded body) and maintains a coherent set of discriminative positive concepts while continuing to acquire informative negative (NOT) concepts as new phases arrive. Fig. A3 shows concept contributions for the same Sturgeon example as in Fig. 6 on ImageNet-Subset under the same 50% concept availability. The dominant contributing concepts remain visually aligned with the input image and follow a similar reasoning pattern to the full-concept setting.

Overall, even under a substantial reduction in concept availability (50% masking), CI-CBM preserves both global interpretability (a stable class-level weight structure across phases) and local interpretability (instance-level concept attributions aligned with the input image). The main qualitative change is that when fine-grained, class-defining attributes are missing from the available concept pool, the explanations shift to closely related, visually grounded alternatives; interpretability is expected to degrade more noticeably only when the concept set becomes so small that such class-defining attributes are systematically unavailable.

## A7 Robustness to Noise in Image-Concept Alignment

To evaluate the sensitivity of our method to noise in concept supervision, we inject Gaussian noise into the concept activation matrix $P$ at different signal-to-noise ratio (SNR) levels during training. Table A7 (Experiment XIII) reports accuracy for SNR levels of 10 dB, 5 dB, and 0 dB. We observe that performance degrades only marginally as noise increases, indicating that the model remains robust even under noisy concept activations.

|  | CIFAR-100 | | | | TinyImageNet | | | |
|---|---|---|---|---|---|---|---|---|
| **SNR Level** | None | 10 dB | 5 dB | 0 dB | None | 10 dB | 5 dB | 0 dB |
| Accuracy (%) | 68.8 | 68.6 | 68.1 | 67.9 | 48.6 | 48.3 | 48.2 | 47.4 |

Table A7: **Experiment XIII (Ablation Study) -** Accuracy vs. SNR in image-concept alignment.

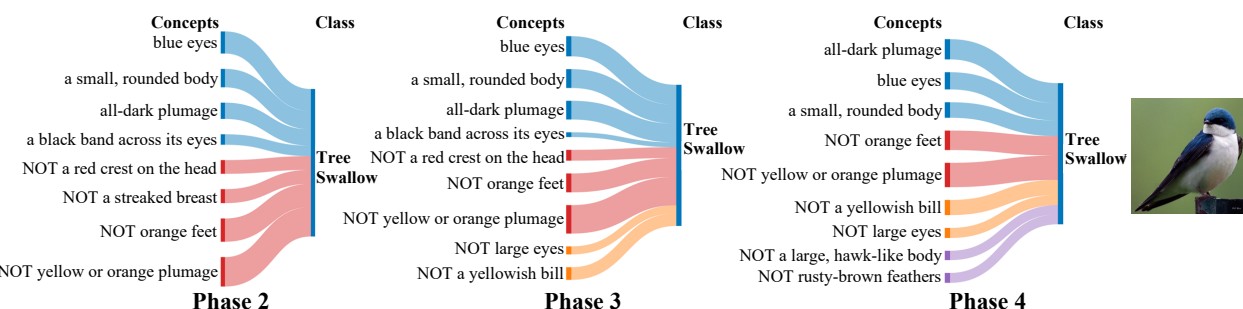

Figure A2: Visualization of the final-layer weights with absolute values greater than 0.2 for the Tree Swallow class in the CUB dataset under a four-phase scenario when training with only 50% of the available concepts (randomly masked), following the same setup as Fig. 1. Concepts with negative weights are labeled as "NOT" concepts. Positive and negative concepts in phase 2 are shown in blue and red, respectively, while concepts added in phases 3 and 4 are shown in orange and purple. Although some class-specific concepts (e.g., blue head and small, forked tail) may be missing due to reduced concept availability, CI-CBM compensates by selecting semantically related cues (e.g., blue eyes and small, rounded body) and maintains a coherent set of discriminative positive and negative features across phases. The thickness of each edge corresponds to the absolute value of the weight.

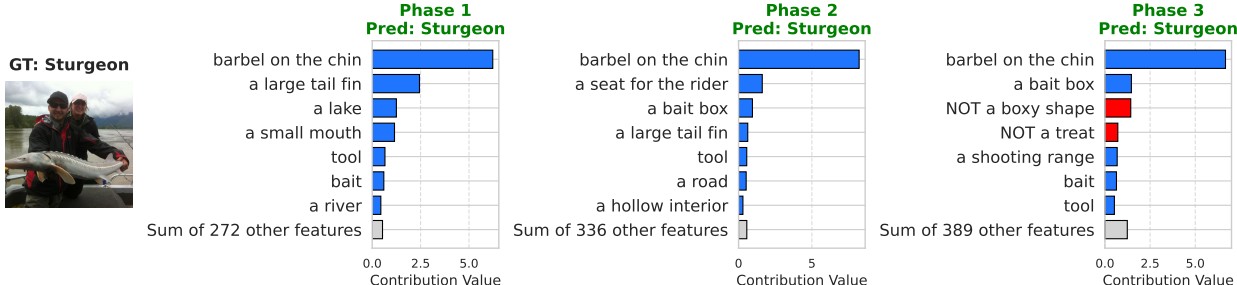

Figure A3: Visualization of model reasoning and concept contributions for the same Sturgeon example as Fig. 6, but trained with only 50% of the available concepts (randomly masked). Despite reduced concept availability, CI-CBM still predicts the correct class and assigns highest contribution to concepts that remain visually aligned with the input.

## A8    Alternative Approach for Learning the Prediction Layer

In addition to CI-CBM, we evaluate two alternative strategies for learning the prediction layer (Table A8, Experiment XIV). **(i) Local Class Discrimination** expands the prediction layer to accommodate new classes at each phase while freezing the weights associated with previously learned classes. This strategy prioritizes separating newly introduced classes from earlier ones. It performs well for $T=5$, where each phase introduces enough classes to reduce overlap; however, performance drops substantially for $T=20$, where fewer classes are added per phase and inter-phase overlap becomes more severe. In contrast, CI-CBM maintains global separability by generating pseudo-concepts, resulting in robust performance across different values of $T$. **(ii) Concept-Space Prototype Generation** tests a natural variant of our pseudo-sample mechanism: instead of generating pseudo-features in the backbone feature space and projecting them through the current concept bottleneck, we attempt to perform the same translation directly in concept space using class prototypes. Specifically, for each previous class we compute a concept-space centroid using the bottleneck representation before fine-tuning on the current phase. Since newly introduced concept dimensions did not exist in earlier phases, we set the corresponding entries of previous-class prototypes to zero and then apply prototype translation in concept space. This variant performs poorly across settings (Table A8), supporting our hypothesis that concept-space translation is unreliable under concept expansion and continual bottleneck adaptation: as phases progress, the concept bottleneck both expands (introducing new concept coordinates)

and drifts (shifting the representation of previously learned concepts under alignment/distillation), causing old class centroids to become misaligned with the updated concept basis. By contrast, CI-CBM generates pseudo-samples in a more stable representation space (the backbone feature space) and then maps them through the current bottleneck to obtain pseudo-concepts, yielding robust performance across different values of $T$.

| Method | CIFAR-100 | | | TinyImageNet | | |
|---|---|---|---|---|---|---|
| | $T{=}5$ | $T{=}10$ | $T{=}20$ | $T{=}5$ | $T{=}10$ | $T{=}20$ |
| Local Class Discrimination | 63.5 | 55.2 | 42.6 | 46.4 | 41.3 | 33.2 |
| Concept-Space Prototype Generation | 48.2 | 35.8 | 27.2 | 33.4 | 25.3 | 18.9 |
| CI-CBM | **68.8** | **68.8** | **67.8** | **48.6** | **48.7** | **48.5** |

Table A8: **Experiment XIV (Ablation Study) -** Performance comparison of alternative prediction layer learning strategy.

| Method | Interpretability | CIFAR-100 ($T{=}60$) | TinyImageNet ($T{=}100$) | ImageNet-Subset ($T{=}60$) |
|---|---|---|---|---|
| FeTrIL (Petit et al., 2023) | ✗ | 59.8 | 50.2 | 65.4 |
| DSLDA (Hayes & Kanan, 2020) | ✗ | 60.5 | 52.6 | 63.6 |
| CI-CBM (ours) | ✓ | 55.9 | 44.8 | 60.8 |
| Full rehearsal | ✓ | 62.0 | 50.1 | 69.3 |

Table A9: **Experiment XV -** Comparison of the average incremental accuracy of CI-CBM and FeTrIL (Petit et al., 2023) and DSLDA (Hayes & Kanan, 2020) in a setting where each incremental phase introduces one new class. This setting represents a special case of Table 1.

## A9    Unique Concept Expansion

Our approach also ensures that only unique concepts are added during each new phase of learning. When a new phase arrives, concepts for the newly introduced classes are generated using GPT-3. However, due to possible similarities between some new concepts and existing classes, naively adding all generated concepts to the concept set can lead to multiple versions of the same concept in the set. Unlike IN2 (Yang et al., 2024), which incorporates duplicate concepts, we ensure that only truly new concepts are added to prevent this redundancy. Figure A4 (Experiment XVI) illustrates the difference in the number of concepts per seen class when duplication is avoided (Unique Concept Set Expansion) versus when all generated concepts are added (Cumulative Concept Count). As shown, avoiding duplication results in nearly half the number of concepts by the final phase. This leads to a much lighter concept bottleneck layer and prediction layer, enabling more efficient optimization. Importantly, removing duplicates improves interpretability by preventing attribution fragmentation: if semantically identical (or near-identical) concepts appear multiple times, their contributions can be split across duplicate dimensions, making explanations longer and less concise. By keeping a single instance of each concept, we preserve the same semantic signal while concentrating weight and attribution on one concept dimension, yielding cleaner and more stable explanations.

## A10    One-Class Increments in EFCIL

Table A9 (Experiment XV) presents results for one-class increments, a task that many EFCIL methods struggle with, as they typically require at least two classes per increment to effectively update the model. The table reports results for CIFAR-100 and ImageNet-Subset with an initial phase of 40 classes and 60 incremental phases, and for TinyImageNet with an initial phase of 100 classes and 100 incremental phases.

## A11    Concept Generation Pipeline

For each new class, we generate candidate concepts using a fixed set of language prompts to query GPT-3. Following the design of LF-CBM (Oikarinen et al., 2023), we apply the following three prompts:

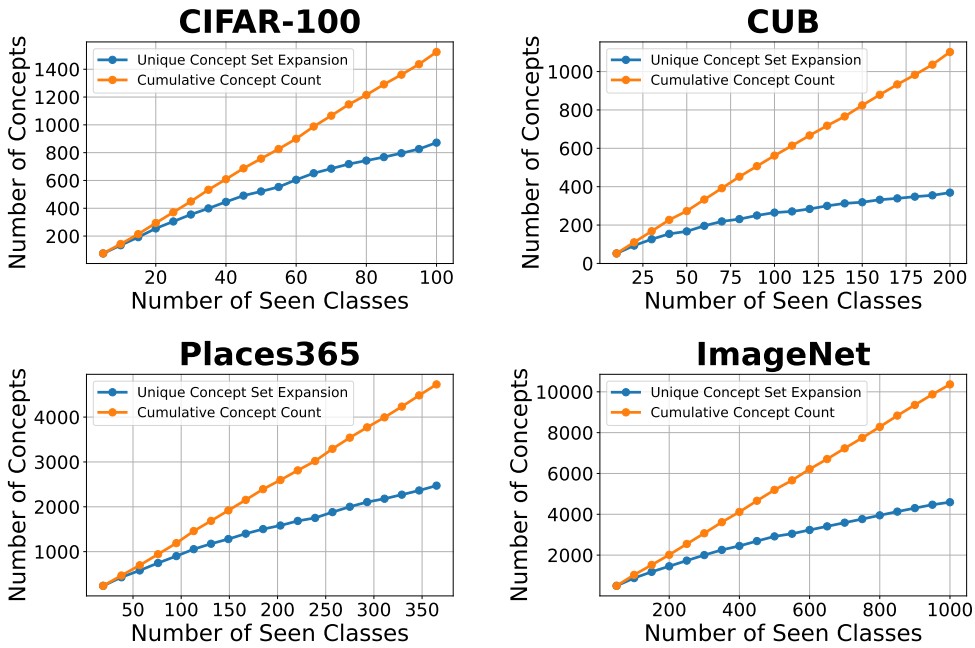

Figure A4: **Experiment XVI (Ablation Study) -** Comparison of the number of concepts used per number of seen classes across different datasets for $T = 20$ phases.

- "List the most important features for recognizing something as a {class}."

- "List the things most commonly seen around a {class}."

- "Give superclasses for the word {class}."

Each prompt is issued twice to increase diversity, and responses are combined to form an initial concept pool. To improve consistency, we use few-shot prompting with two fixed example classes and their expected outputs. These examples are shared across all datasets and phases.

The raw concept set is then filtered in three stages:

1. Length filter: Remove concepts longer than 30 characters.

2. Class similarity filter: Remove concepts with cosine similarity > 0.85 to any class name, using Sentence-Transformer and CLIP text embeddings.

3. Redundancy filter: Remove near-duplicate concepts with cosine similarity > 0.9 to any earlier concept in the set.

This filtered set is then used to update the concept set. All steps are automated and applied incrementally for each new phase. The full pipeline is implemented in the released codebase.

## A12 Impact of Distillation Regularizer Weight on Accuracy.

To quantify the effect of the distillation regularizer, $\beta$, in Eq. 3 on class-incremental performance, we sweep $\beta \in \{0, 0.25, 0.5, 1, 2, 5\}$ and report the resulting average incremental accuracy. For CIFAR-100 and TinyImageNet, we follow the pretrained-backbone setting of Experiment I (Table 1); for ImageNet-Subset, we follow the non-pretrained setting of Experiment II (Table 2). Table A10 shows that introducing the distillation

term ($\beta > 0$) consistently improves accuracy over the no-distillation baseline ($\beta = 0$), indicating that distillation effectively mitigates concept drift across phases. Performance typically peaks at a moderate value, with $\beta = 1$ achieving the best overall accuracy across datasets. We therefore set $\beta = 1$ in all experiments.

| Concept reg | CIFAR-100 | | | TinyImageNet | | | ImageNet Subset | | |
|---|---|---|---|---|---|---|---|---|---|
| | $T=5$ | $T=10$ | $T=20$ | $T=4$ | $T=10$ | $T=20$ | $T=5$ | $T=10$ | $T=20$ |
| $\beta = 0$ | 68.4 | 68.2 | 67.6 | 48.2 | 48.3 | 48.2 | 66.5 | 64.9 | 66.1 |
| $\beta = 0.25$ | 68.5 | 68.6 | 67.5 | 48.2 | 48.7 | 48.4 | 67.3 | 65.5 | 66.8 |
| $\beta = 0.5$ | 68.6 | 68.7 | 67.7 | **48.6** | **48.8** | 48.6 | 67.4 | 65.8 | 66.7 |
| $\beta = 1$ | **68.8** | **68.8** | **67.8** | **48.6** | 48.7 | 48.5 | **67.8** | **66.2** | **66.9** |
| $\beta = 2$ | 68.7 | 68.6 | **67.8** | 48.5 | 48.7 | **48.7** | 67.3 | **66.2** | 66.4 |
| $\beta = 5$ | 67.8 | 67.8 | 67.2 | 48.4 | 48.7 | 48.5 | 67.2 | **66.2** | 66.4 |

Table A10: **Experiment XVII (Ablation) – Effect of distillation weight $\beta$.** Average incremental accuracy (%) when varying the distillation regularizer weight $\beta$ in Eq. 3. CIFAR-100 and TinyImageNet use the pretrained-backbone protocol of Experiment I, while ImageNet-Subset uses the non-pretrained protocol of Experiment II. Results are reported for different numbers of incremental phases $T$.

## A13 Impact of the Distillation Regularizer on Concept Fidelity.

To assess the effect of the distillation regularizer in Eq. 3 on interpretability and concept fidelity of the concept bottleneck, we analyze the learned concept representations on the test split of each benchmark. Let the test set be denoted by $D_{\text{test}} = \{x_1, \ldots, x_N\}$ and the final concept vocabulary by $C_T = \{t_1, \ldots, t_M\}$. We first compute and store a CLIP-based concept-activation matrix $P \in \mathbb{R}^{N \times M}$, where each entry measures the alignment between the $i$-th test image and the $j$-th concept text via CLIP embeddings:

$$P_{i,j} = E_I(x_i) \cdot E_T(t_j), \tag{A5}$$

with $E_I$ and $E_T$ denoting the CLIP image and text encoders, respectively. Next, we record the activations of the concept-bottleneck (target) neurons on the same test set: for each bottleneck unit $k$ and each image $x_i \in D_{\text{test}}$, we compute the scalar activation $A_k(x_i)$. This yields an activation vector for neuron $k$,

$$q_k = [A_k(x_1), \ldots, A_k(x_N)]^\top \in \mathbb{R}^N. \tag{A6}$$

Given neuron $k$, we assign it a concept label by comparing $q_k$ against the concept-specific response profiles induced by $P$, selecting the most similar concept $t_m$ under cosine similarity. Concretely, letting $p_m = P_{:,m} \in \mathbb{R}^N$ denote the $m$-th column of $P$ (the activation profile of concept $t_m$ over $D_{\text{test}}$), we define

$$\hat{m}(k) = \arg \max_{m \in \{1, \ldots, M\}} \cos(q_k, p_m), \tag{A7}$$

and associate unit $k$ with concept $t_{\hat{m}(k)}$, indicating which semantic concept the unit responds to most strongly. We follow the same protocol as Experiment XVII: CIFAR-100 and TinyImageNet use a pretrained backbone (Experiment I), while ImageNet-Subset uses the non-pretrained setting (Experiment II). Table A11 reports two complementary concept-fidelity metrics. First, CLIP cosine similarity measures the cosine similarity between the predicted concept and the ground-truth concept in the CLIP text-embedding space. Second, Top-5 concept accuracy measures whether the ground-truth concept appears among the top-5 most similar concepts ranked by cosine similarity. For reference, the number of candidate concepts is $M = 872$ for CIFAR100, $M = 1700$ for TinyImageNet, and $M = 979$ for ImageNet Subset. As shown, enabling the distillation regularizer consistently improves concept-fidelity measures across datasets, indicating that it better preserves semantic alignment of concept units while learning incrementally. This supports our hypothesis that regularizing against drift in previously learned concepts is essential for maintaining interpretable and stable concept representations under continual adaptation.

| Metric | Concept reg | CIFAR100 | | | TinyImageNet | | | ImageNet Subset | | |
|---|---|---|---|---|---|---|---|---|---|---|
| | | $T$=5 | $T$=10 | $T$=20 | $T$=5 | $T$=10 | $T$=20 | $T$=5 | $T$=10 | $T$=20 |
| CLIP cosine similarity | ✗ | 0.916 | 0.891 | 0.874 | 0.876 | 0.864 | 0.854 | 0.840 | 0.822 | 0.786 |
| | ✓ | **0.927** | **0.929** | **0.890** | **0.883** | **0.886** | **0.869** | **0.914** | **0.887** | **0.850** |
| Top-5 accuracy | ✗ | 84.7 | 76.5 | 68.8 | 70.8 | 65.0 | 61.4 | 54.3 | 41.6 | 23.6 |
| | ✓ | **87.7** | **87.0** | **71.8** | **74.7** | **74.0** | **66.5** | **80.6** | **74.9** | **57.7** |

Table A11: **Experiment XVIII (Ablation Study) -** Impact of the distillation-based concept regularizer on concept fidelity of bottleneck units across continual-learning phase granularities $T \in \{5, 10, 20\}$. We report *CLIP cosine similarity* between the predicted and ground-truth concepts in the CLIP text-embedding space, and *Top-5 concept accuracy* (whether the ground-truth concept appears among the top-5 most similar concepts under cosine similarity). Results are shown for CIFAR100, TinyImageNet, and ImageNet Subset. Enabling concept regularization (✓) consistently improves semantic alignment compared to no regularization (✗), indicating reduced drift of learned concept representations during incremental adaptation.

## A14    Impact of Vision-Language Model (SigLIP vs. CLIP).

We compute the concept-activation matrix $P$ using a vision-language model (VLM) to align concept-bottleneck units with their corresponding concepts. In this ablation, we keep the full training pipeline identical to Experiment I (Table 1) and only swap the VLM used to compute $P$ (CLIP vs. SigLIP). Table A12 shows that replacing CLIP with SigLIP consistently improves class-incremental accuracy across datasets and phase configurations, suggesting stronger concept alignment in $P$ and, consequently, better downstream performance. Beyond accuracy, we also evaluate concept fidelity using the same metrics and

| Method | CIFAR-100 | | | CUB | | | TinyImageNet | | |
|---|---|---|---|---|---|---|---|---|---|
| | $T$=5 | $T$=10 | $T$=20 | $T$=4 | $T$=10 | $T$=20 | $T$=5 | $T$=10 | $T$=20 |
| CI-CBM (CLIP) | 68.6 | 68.7 | 67.2 | 60.5 | 62.4 | 63.6 | 47.0 | 47.2 | 47.3 |
| CI-CBM (SigLIP) | **68.8** | **68.8** | **67.8** | **62.2** | **65.3** | **66.1** | **48.6** | **48.7** | **48.5** |

Table A12: **Experiment XIX (Ablation Study) -** Incremental accuracy (%) when computing the concept activation matrix $P$ with different VLMs (SigLIP vs. CLIP). This experiment follows the same setup as Experiment I (Table 1) and only swaps the VLM used to compute $P$.

evaluation protocol introduced in Section A13. Table A13 shows that SigLIP improves concept-fidelity measures as well, indicating that the learned concept units are more semantically aligned and interpretable when $P$ is computed with SigLIP.

| Metric | VLM | CIFAR100 | | | CUB | | | TinyImageNet | | |
|---|---|---|---|---|---|---|---|---|---|
| | | $T$=5 | $T$=10 | $T$=20 | $T$=4 | $T$=10 | $T$=20 | $T$=5 | $T$=10 | $T$=20 |
| CLIP cosine similarity | CLIP | 0.890 | 0.888 | 0.890 | 0.949 | 0.854 | 0.830 | 0.866 | 0.866 | 0.845 |
| | SigLIP | **0.927** | **0.929** | **0.837** | **0.961** | **0.876** | **0.837** | **0.883** | **0.886** | **0.869** |
| Top-5 accuracy | CLIP | 73.6 | 74.9 | 49.4 | 72.9 | 58.5 | 53.4 | 66.6 | 66.5 | 57.0 |
| | SigLIP | **87.7** | **87.0** | **71.8** | **78.9** | **68.8** | **49.4** | **74.7** | **74.0** | **66.5** |

Table A13: **Experiment XX (Ablation Study) -** Concept-fidelity metrics when computing the concept-activation matrix $P$ with different VLMs (CLIP vs. SigLIP), while keeping the training protocol identical to Experiment I (Table 1). We report CLIP cosine similarity and Top-5 concept accuracy using the same evaluation procedure as Section A13. SigLIP yields higher concept-fidelity scores in most settings, indicating improved semantic alignment of the learned concept units.

## A15 Pseudo-Features Track Real Feature Geometry Under a Prototype Classifier

The pseudo-feature construction is designed to approximate the feature distribution of previously seen classes using only summary statistics and the current feature space. To assess whether pseudo-features behave similarly to real samples in a realistic, multi-class setting, we perform a controlled analysis using a simple cosine-prototype classifier.

We consider a class-incremental protocol with $T = 10$ phases on CIFAR100. Given a frozen feature extractor, we compute a centroid (prototype) for each class using its *training* samples, and classify by assigning each feature to the class with the maximum cosine similarity to its centroid. We report results for two backbone settings: (*i*) an ImageNet-pretrained ResNet-18 (corresponding to Experiment I in Table 1), and (*ii*) the same architecture trained on the first-phase data only and then frozen (corresponding to Experiment II in Table 2).

Fig. A5 visualizes four phase-by-phase matrices. For each entry $(i, j)$, we evaluate performance on classes introduced at phase $j$ after learning up to phase $i$:

- Training (cosine-prototype): accuracy on training samples using the phase-$i$ centroids.

- Pseudo-feature (cosine-prototype): for each class introduced at phase $j < i$, we generate pseudo-features at phase $i$ and report the fraction whose nearest centroid (among all phase-$i$ centroids) is their own class centroid.

- Test (cosine-prototype): accuracy on held-out real samples using the phase-$i$ centroids.

- Ours (test): test accuracy of the proposed method (not the cosine-prototype classifier).

Across phases, the pseudo-feature matrix closely matches the corresponding training and test matrices under the cosine-prototype classifier. This indicates that, when evaluated by the same nearest-centroid decision rule, pseudo-features induce similar confusion patterns and decision boundaries to those produced by real data.

We further analyze a second setting in which the backbone is *not* ImageNet-pretrained, but instead trained only on the first-phase data and then kept fixed. In this scenario, we observe a qualitatively similar alignment: pseudo-feature accuracy remains close to the accuracy measured on real training and test samples under the same cosine-prototype classifier. At the same time, the accuracy is noticeably higher for classes introduced in the first phase, while it is consistently lower for classes introduced in later phases across all three matrices (training, pseudo-feature, and test). A plausible explanation is that, the backbone is trained only using first-phase supervision, which shapes the backbone's feature space to separate the first-phase classes well. As a result, features from early classes tend to cluster more tightly around their class centroids. In contrast, for classes introduced in later phases (which did not participate in backbone training), their features may be less clustered and less centered around a single centroid, making nearest-centroid classification less reliable for these classes.

Importantly, this trend should not be attributed to pseudo-features alone. Instead, it reflects a limitation of centroid-based modeling when the backbone feature space is not well clustered for newly introduced classes: in that case, nearest-centroid accuracy drops for both real features and pseudo-features at the same time. This also highlights the role of representation quality: stronger pretrained backbones typically provide better class separation and therefore improve both prototype classification and pseudo-feature reliability. However, relying solely on strong pretraining does not address the core objective of class-incremental learning, namely acquiring new knowledge beyond what is already encoded in the backbone.

Despite these representation constraints, our full method improves test accuracy in both the pretrained and first-phase-trained settings, with particularly pronounced gains in the latter. This suggests that the concept bottleneck layer provides an additional, structured space in which samples become more separable, even when the raw feature space is not well organized for newly introduced classes. In addition to improving incremental generalization, this mechanism also supports concept-based interpretability by grounding predictions in a small set of activated concepts.

## A16 Additional Visualization for Model Reasoning

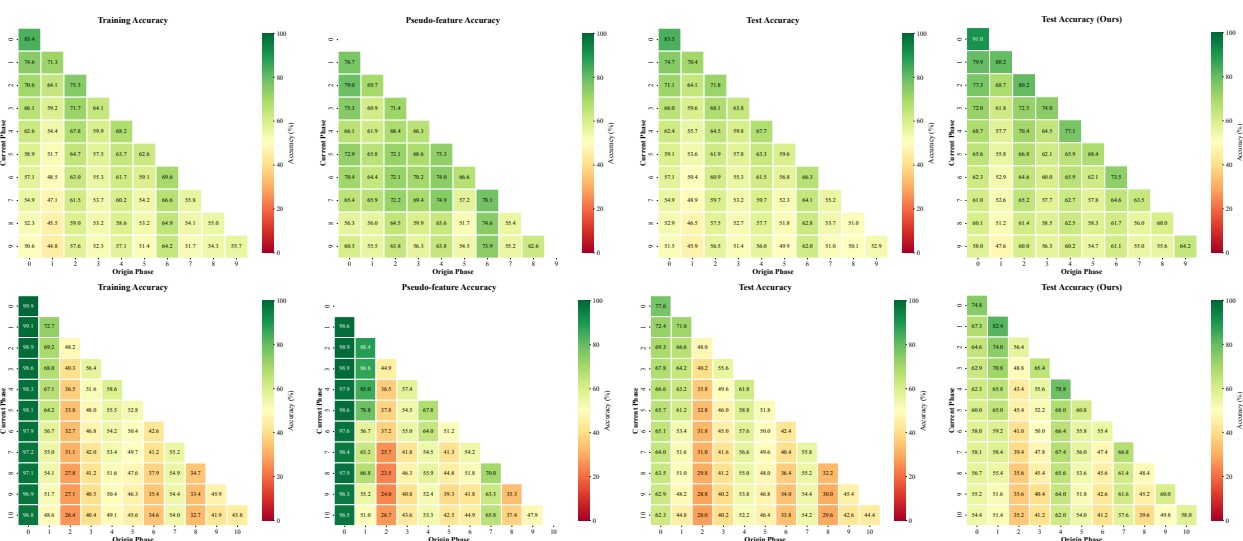

Figure A5: **Experiment XXI -** We visualize phase-by-phase accuracy matrices with entry $(i, j)$ evaluating classes introduced at phase $j$ after learning up to phase $i$. Columns 1–3 use the same cosine-prototype classifier (nearest class centroid in cosine similarity): (1) training accuracy on real training samples, (2) pseudo-feature accuracy (fraction of pseudo-features whose nearest centroid is their own class centroid), and (3) validation/test accuracy on real held-out samples. (4) reports test accuracy of *our method* (not the cosine-prototype classifier). Top row: ImageNet-pretrained ResNet-18 backbone. Bottom row: ResNet-18 trained only on first-phase data. Pseudo-feature accuracy closely tracks real-data accuracy under the same cosine-prototype rule, while our method substantially improves incremental test performance in the weaker-backbone setting.

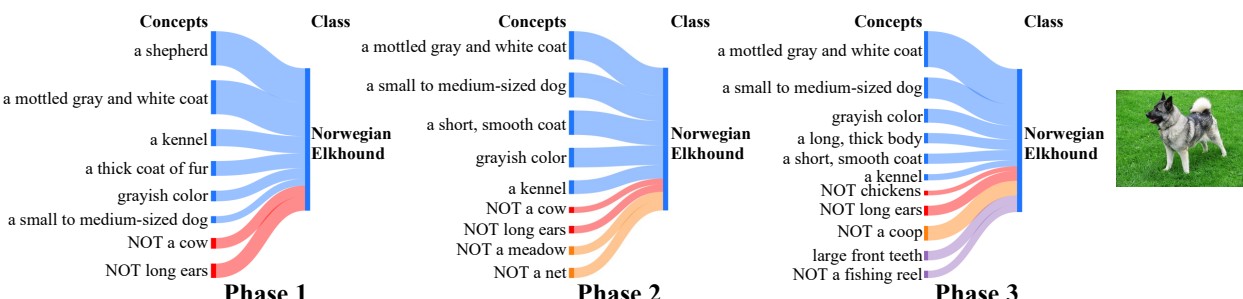

Figure A6: Visualization of the final layer weights with absolute values greater than 0.2 for the **Norwegian Elkhound** class in the ImageNet-Subset dataset under a five-phase scenario. Concepts with negative weights are labeled as "NOT" concepts. Positive and negative concepts in phase 1 are shown in blue and red, respectively, while concepts added in phases 2 and 3 are shown in orange and purple. The thickness of each edge corresponds to the absolute value of the weight. (Complementary to Figure 1)

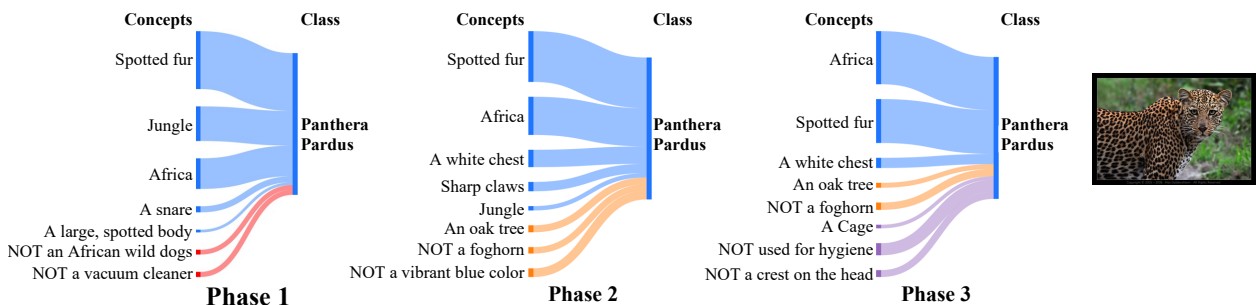

Figure A7: Visualization of the final layer weights with absolute values greater than 0.2 for the **Panthera Pardus** class in the ImageNet-Subset dataset under a five-phase scenario. Concepts with negative weights are labeled as "NOT" concepts. Positive and negative concepts in phase 1 are shown in blue and red, respectively, while concepts added in phases 2 and 3 are shown in orange and purple. The thickness of each edge corresponds to the absolute value of the weight. (Complementary to Figure 1)

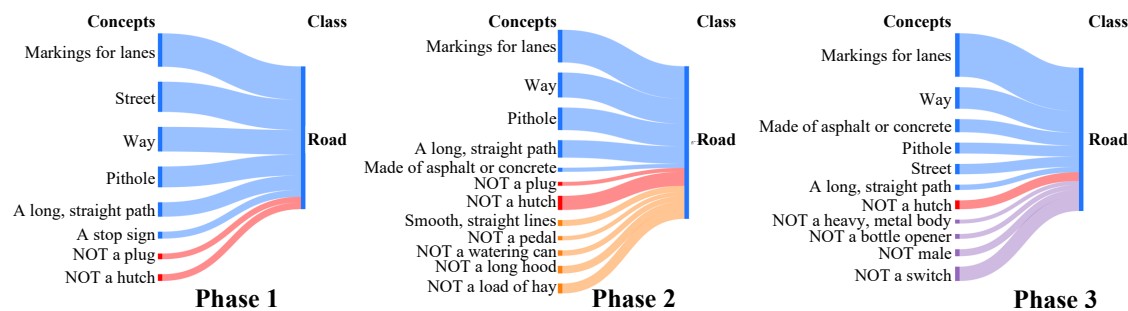

Figure A8: Visualization of the final layer weights with absolute values greater than 0.2 for the **Road** class in the CIFAR-100 dataset under a five-phase scenario. Concepts with negative weights are labeled as "NOT" concepts. Positive and negative concepts in phase 1 are shown in blue and red, respectively, while concepts added in phases 2 and 3 are shown in orange and purple. The thickness of each edge corresponds to the absolute value of the weight. (Complementary to Figure 1)

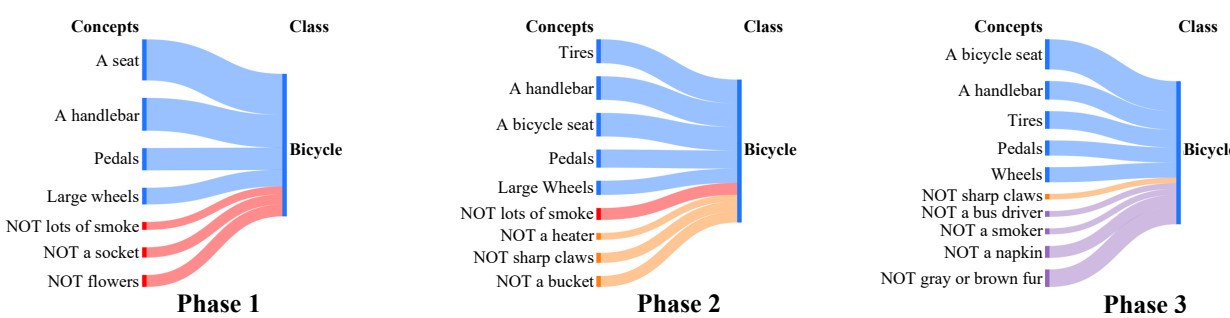

Figure A9: Visualization of the final layer weights with absolute values greater than 0.2 for the **Bicycle** class in the CIFAR-100 dataset under a five-phase scenario. Concepts with negative weights are labeled as "NOT" concepts. Positive and negative concepts in phase 1 are shown in blue and red, respectively, while concepts added in phases 2 and 3 are shown in orange and purple. The thickness of each edge corresponds to the absolute value of the weight. (Complementary to Figure 1)

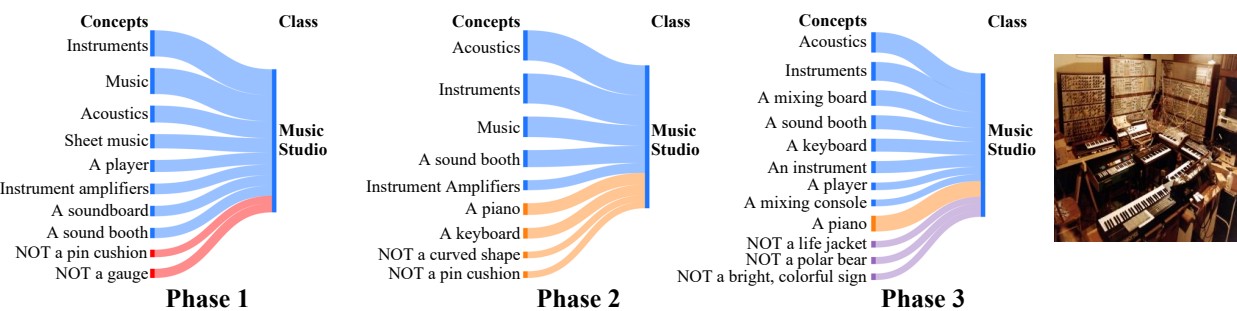

Figure A10: Visualization of the final layer weights with absolute values greater than 0.2 for the **Music Studio** class in the Places365 dataset under a five-phase scenario. Concepts with negative weights are labeled as "NOT" concepts. Positive and negative concepts in phase 1 are shown in blue and red, respectively, while concepts added in phases 2 and 3 are shown in orange and purple. The thickness of each edge corresponds to the absolute value of the weight. (Complementary to Figure 1)

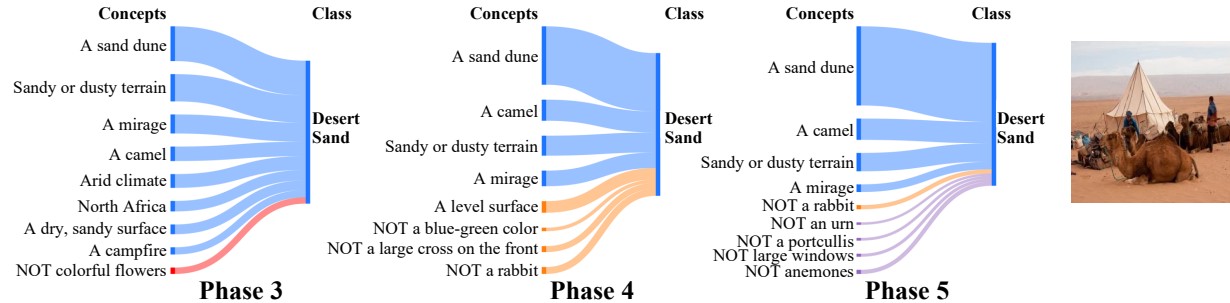

Figure A11: Visualization of the final layer weights with absolute values greater than 0.2 for the **Desert (Sand)** class in the Places365 dataset under a five-phase scenario. Concepts with negative weights are labeled as "NOT" concepts. Positive and negative concepts in phase 3 are shown in blue and red, respectively, while concepts added in phases 4 and 5 are shown in orange and purple. The thickness of each edge corresponds to the absolute value of the weight. (Complementary to Figure 1)

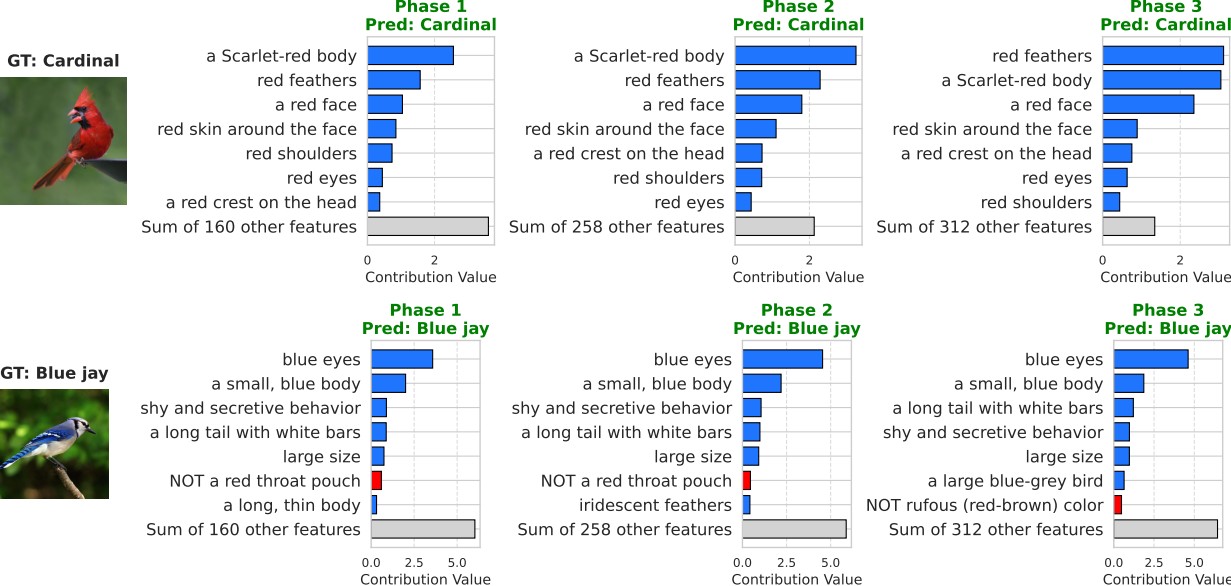

Figure A12: Visualization of model reasoning and concept contributions for images of the Cardinal and Blue Jay classes, introduced in the first phase of the CUB dataset. (Complementary to Figure 6)

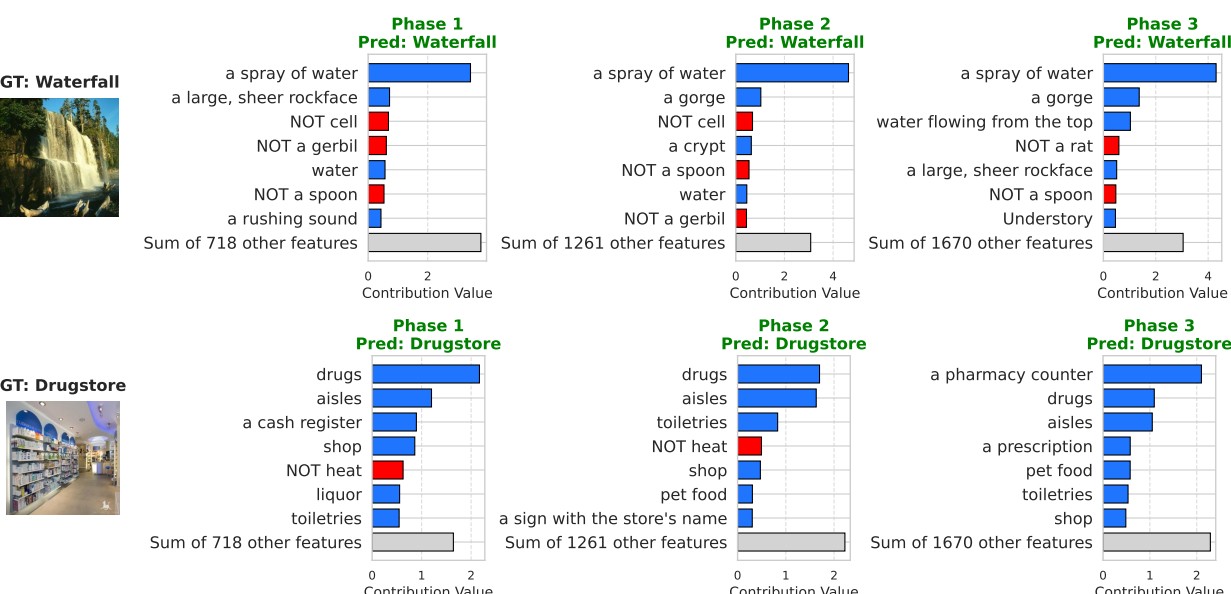

Figure A13: Visualization of model reasoning and concept contributions for images of the Waterfall and Drugstore classes, introduced in the first phase of the Places365 dataset. (Complementary to Figure 6)

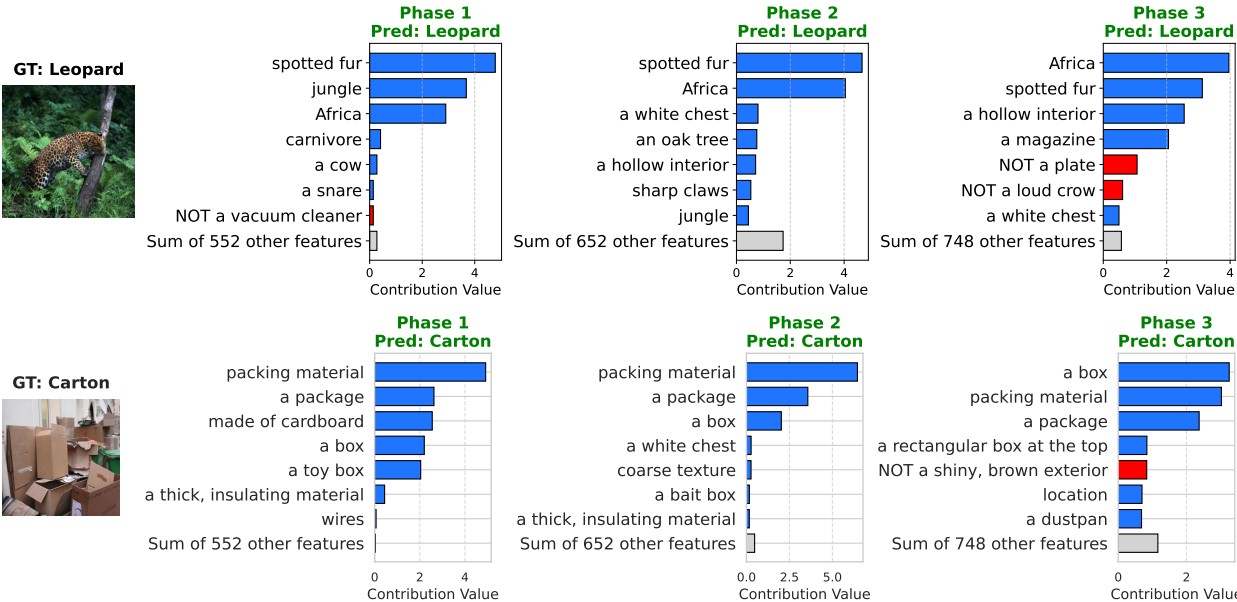

Figure A14: Visualization of model reasoning and concept contributions for images of the Leopard and Carton classes, introduced in the first phase of the ImageNet-Subset. (Complementary to Figure 6)

