# OpenReview forum: "CI-CBM: Class-Incremental Concept Bottleneck Model for Interpretable Continual Learning"
_TMLR — Accepted by TMLR_

### Review · Reviewer_oTPr · 2025-12-14

**Summary Of Contributions:**

In this paper, the authors study the problem of interpretable class-incremental learning (CIL). To be specific, they are interested in designing a new concept-bottleneck-layer which can incrementally acquire new concepts to guide CIL. The authors introduce three modules: Concept set expansion, Concept bottleneck layer distillation, Dynamic adaptation of the prediction layer. The proposed method is compared against interpretable CIL methods as well as non-interpretable CIL methods. Significant improvements over interpretable CIL methods are reported.

Strengths:

+ Designing interpretable CIL approaches is an important problem in ML.
+ Novel approach, bringing together existing ideas and tricks from the literature.
+ Significant gains are reported.
+ Overall, well-written and easy-to-follow text.

**Audience:**

Yes

**Audience Explanation:**

Interpretable class-incremental learning is of interest to the machine learning community.

**Broader Impact Concerns:**

N/A.

**Claims And Evidence:**

Yes

**Claims Explanation:**

The paper includes comparisons with the alternative approaches and reports significant improvements.

**Requested Changes:**

Weaknesses:

1. Some design choices or steps have not been justified:

1.1. Module 1:

1.1.1. "If concepts related to the new classes already exist due to their similarity to previous classes, they are not added again." => Please describe how you calculate similarity between concepts.

1.1.2. "In our experiments, we used SigLIP (Zhai et al., 2023) instead of CLIP (Radford et al., 2021), which is used in LF-CBM (Oikarinen et al., 2023), as the multimodal model for computing Pt. SigLIP is a recent model that focuses on image-text pairs and employs a sigmoid loss function, in contrast to the softmax-based contrastive learning approach used in CLIP." => Please provide justification for why you used SigLIP instead of CLIP.

1.1.3. Please add here a short explanation for why P_{t-1} is not used while calculating P_t.

1.2. Module 2:

1.2.1. Please include an ablation on the contribution of distillation loss, by introducing a weight.

1.3. Theoretical Perspective => This is a toy-example / controlled-analysis rather than a theoretical analysis.

1.4. It would be helpful to add an algorithm to describe all steps of the algorithm.

2. Experiments:

2.1. Table 1: The proposed method's performance on the CUB dataset increases significantly with the increasing number of tasks. It would be helpful to explain this counter-intuitive result.

2.2. Figure 6: Such visualizations are very helpful. However, the plot clearly shows that some positive concepts (e.g., a large tail fin) disappear and some positive concepts (e.g., "a fish") diminish in the new phases. The remaining concepts turned out to be sufficient in such a case but this example shows that the proposed method is not able to preserve important concepts through phases.

2.3. The proposed method is only compared to the other interpretable methods (ICICLE, IN2) in Table 1. Not sure why the remaining experiments (e.g., Experiment 3) do not include results by ICICLE and IN2.


Minor comments:
- "Cos cubed" => "cosine-cubed"?
- The following guide can be helpful in writing equations:
 https://wp.optics.arizona.edu/kupinski/wp-content/uploads/sites/91/2023/05/MerminEquations.pdf
- Eq 1: Please also explain what L and q denote.
- "the first to the last phase T," => "the first to the last phase $T$,"
- "prediction layer W_F^{t-1}" => Either include b_F here as well or modify Eq 2 to absorb the bias in W_F.
- "Following (Petit et al., 2023)," => "Following Petit et al. (2023),"
- "(Zhao et al.)," => Year is missing.

---

> ### Author Response · Authors · 2026-02-22
> **Response to Reviewer oTPr**
>
> We are grateful to the reviewer for the positive and constructive review. Below we respond point-by-point and summarize the corresponding updates in the revised manuscript.
>
> ### 1. Design choices and clarifications
>
> **Concept similarity criterion in Module (I).**
> This mechanism was already described in the original submission (Appendix A11: Concept Generation Pipeline): for each newly generated concept, we compute a text embedding (using a text encoder such as Sentence-Transformer or CLIP) and remove the concept if its cosine similarity to any previously stored class name or concept exceeds a fixed threshold. In the revised manuscript, we made the following changes: (i) added a clearer explanation directly in Module (I) (Incremental Concept Set Expansion), and (ii) added an explicit reference to Appendix A11 for the complete filtering pipeline and thresholds.
>
> **Why SigLIP instead of CLIP for computing $P_t$.**
> Following your suggestion, we added a dedicated ablation experiment in Appendix A14: Impact of Vision-Language Model (SigLIP vs. CLIP). The training pipeline is kept identical to our main interpretable setting, and we only swap the VLM used to compute $P$ (CLIP vs. SigLIP). The results show that using SigLIP yields consistently better incremental accuracy across datasets and phase configurations, suggesting improved concept alignment in $P$ and better downstream performance.
> | Method | CIFAR-100 (T=5) | CIFAR-100 (T=10) | CIFAR-100 (T=20) | CUB (T=4) | CUB (T=10) | CUB (T=20) | TinyImageNet (T=5) | TinyImageNet (T=10) | TinyImageNet (T=20) |
> |---|---:|---:|---:|---:|---:|---:|---:|---:|---:|
> | CI-CBM (CLIP)  | 68.6 | 68.7 | 67.2 | 60.5 | 62.4 | 63.6 | 47.0 | 47.2 | 47.3 |
> | CI-CBM (SigLIP) | **68.8** | **68.8** | **67.8** | **62.2** | **65.3** | **66.1** | **48.6** | **48.7** | **48.5** |
>
> In addition, we also report concept-fidelity metrics under the same swap. SigLIP improves concept fidelity in most settings, indicating that the learned bottleneck units are more semantically aligned with the intended concepts when $P$ is computed via SigLIP.
> | Metric | VLM | CIFAR100 (T=5) | CIFAR100 (T=10) | CIFAR100 (T=20) | CUB (T=4) | CUB (T=10) | CUB (T=20) | TinyImageNet (T=5) | TinyImageNet (T=10) | TinyImageNet (T=20) |
> |---|---|---:|---:|---:|---:|---:|---:|---:|---:|---:|
> | CLIP cosine similarity | CLIP | 0.890 | 0.888 | 0.890 | 0.949 | 0.854 | 0.830 | 0.866 | 0.866 | 0.845 |
> | CLIP cosine similarity | SigLIP | **0.927** | **0.929** | **0.837** | **0.961** | **0.876** | **0.837** | **0.883** | **0.886** | **0.869** |
> | Top-5 accuracy | CLIP | 73.6 | 74.9 | 49.4 | 72.9 | 58.5 | 53.4 | 66.6 | 66.5 | 57.0 |
> | Top-5 accuracy | SigLIP | **87.7** | **87.0** | **71.8** | **78.9** | **68.8** | **49.4** | **74.7** | **74.0** | **66.5** |
>
> **Why $P_{t-1}$ is not used to compute $P_t$.**
> In our setting, $P_t \in \mathbb{R}^{N_t \times M_t}$ is defined as the image–concept activation matrix computed only on the current-phase training images $X_t$ and the current concept set $C_t$. In contrast, $P_{t-1}$ is defined over a different image set $X_{t-1}$ and an older concept vocabulary $C_{t-1}$, which does not include the newly added concepts at phase $t$. Therefore, $P_{t-1}$ cannot be directly reused to construct $P_t$. Reusing $P_{t-1}$ would require recomputing similarities between past images and the newly introduced concepts, which is not permitted in the exemplar-free CIL setting (due to storage/privacy constraints and the absence of old data). Hence, we compute $P_t$ from scratch using only $D_t$ and $C_t$ at each phase.

---

> ### Author Response · Authors · 2026-02-22
> **Response to Reviewer oTPr (Part 2)**
>
> **Ablation on distillation loss contribution.**
> We implemented this suggestion by introducing a distillation weight $\beta$ and sweeping $\beta \in \lbrace 0, 0.25, 0.5, 1, 2, 5 \rbrace$. We added two complementary analyses:
>
> **Impact of $\beta$ on class-incremental accuracy.**
> We added Appendix A12: Impact of Distillation Regularizer Weight on Accuracy, reporting average incremental accuracy across multiple datasets and phase granularities. Overall, adding distillation ($\beta>0$) consistently improves accuracy over the no-distillation baseline ($\beta=0$), and performance typically peaks at a moderate value (we select $\beta=1$ for the rest of the paper).
> | Concept reg | CIFAR-100 (T=5) | CIFAR-100 (T=10) | CIFAR-100 (T=20) | TinyImageNet (T=4) | TinyImageNet (T=10) | TinyImageNet (T=20) | ImageNet Subset (T=5) | ImageNet Subset (T=10) | ImageNet Subset (T=20) |
> |---|---:|---:|---:|---:|---:|---:|---:|---:|---:|
> | β = 0    | 68.4 | 68.2 | 67.6 | 48.2 | 48.3 | 48.2 | 66.5 | 64.9 | 66.1 |
> | β = 0.25 | 68.5 | 68.6 | 67.5 | 48.2 | 48.7 | 48.4 | 67.3 | 65.5 | 66.8 |
> | β = 0.5  | 68.6 | 68.7 | 67.7 | **48.6** | **48.8** | 48.6 | 67.4 | 65.8 | 66.7 |
> | β = 1    | **68.8** | **68.8** | **67.8** | **48.6** | 48.7 | 48.5 | **67.8** | **66.2** | **66.9** |
> | β = 2    | 68.7 | 68.6 | **67.8** | 48.5 | 48.7 | **48.7** | 67.3 | **66.2** | 66.4 |
> | β = 5    | 67.8 | 67.8 | 67.2 | 48.4 | 48.7 | 48.5 | 67.2 | **66.2** | 66.4 |
>
> **Impact of distillation on concept fidelity:**
> In the revised manuscript, we added Appendix A13: Impact of the Distillation Regularizer on Concept Fidelity, which provides the full evaluation protocol and implementation details. In our work, concept fidelity measures whether the concept label assigned to a neuron in the concept-bottleneck layer remains faithful to that same semantic concept at the end of continual training—i.e., whether the neuron preserves its intended meaning after learning multiple incremental phases rather than drifting to represent a different concept.
>
> We quantify this using two complementary metrics: (i) CLIP cosine similarity in the CLIP text-embedding space between the predicted and ground-truth concepts, and (ii) Top-5 concept accuracy, which checks whether the ground-truth concept appears among the five most similar concepts ranked by cosine similarity.
>
> As reported in Appendix A13, enabling the distillation-based concept regularizer consistently improves both concept-fidelity measures across datasets and phase granularities, supporting our conclusion that distillation mitigates semantic drift and helps maintain stable, interpretable concept representations throughout incremental learning.
> | Metric | Concept reg | CIFAR100 (T=5) | CIFAR100 (T=10) | CIFAR100 (T=20) | TinyImageNet (T=5) | TinyImageNet (T=10) | TinyImageNet (T=20) | ImageNet Subset (T=5) | ImageNet Subset (T=10) | ImageNet Subset (T=20) |
> |---|:---:|---:|---:|---:|---:|---:|---:|---:|---:|---:|
> | CLIP cosine similarity | ✗ | 0.916 | 0.891 | 0.874 | 0.876 | 0.864 | 0.854 | 0.840 | 0.822 | 0.786 |
> | CLIP cosine similarity | ✓ | **0.927** | **0.929** | **0.890** | **0.883** | **0.886** | **0.869** | **0.914** | **0.887** | **0.850** |
> | Top-5 accuracy | ✗ | 84.7 | 76.5 | 68.8 | 70.8 | 65.0 | 61.4 | 54.3 | 41.6 | 23.6 |
> | Top-5 accuracy | ✓ | **87.7** | **87.0** | **71.8** | **74.7** | **74.0** | **66.5** | **80.6** | **74.9** | **57.7** |

---

> ### Author Response · Authors · 2026-02-22
> **Response to Reviewer oTPr (Part 3)**
>
> ### 2. “Theoretical Perspective” wording and strengthening beyond a toy example
>
> We agree with the reviewer that this is not a formal theoretical result. Following the suggestion, we renamed the section to “Controlled Geometric Perspective.”
>
> We also added an explicit paragraph clarifying the assumption behind this controlled picture: it assumes class-conditional features are reasonably concentrated around their centroids so that mean-shifts preserve relevant geometry. To bridge this controlled analysis to realistic high-dimensional representations, and to assess sensitivity to pretrained vs. non-pretrained feature extractors, we added an empirical validation in Supplementary Section A15 (pseudo-feature reliability under a prototype classifier).
>
> We evaluate whether pseudo-features and real features behave similarly under the same cosine-prototype classifier across phases, under two backbone regimes: ImageNet-pretrained vs. first-phase-trained-and-frozen. We report phase-by-phase accuracy matrices for (i) real training features, (ii) pseudo-features, and (iii) real test features, showing that pseudo-feature accuracy tracks the corresponding real-feature accuracy patterns in both regimes. This supports that pseudo-features capture the same class geometry that a centroid-based classifier would “see” in the backbone feature space, and it makes explicit when this proxy is more reliable (pretrained backbone) versus more constrained (weaker/non-pretrained backbone). Importantly, even under this constraint, our full method still improves test accuracy, indicating that the concept bottleneck provides an additional structured space that helps separability beyond the raw feature geometry.
>
> ### 3. Algorithm description
>
> Thank you for the suggestion. We added Algorithm 1, which provides a complete step-by-step summary across phases and explicitly lists Modules I–III and the training sequence.
>
> ### 4. CUB performance increases with more tasks
>
> We agree this can look counter-intuitive at first. A plausible explanation is that CUB has a highly structured and well-separated feature space under an ImageNet-pretrained backbone, so pseudo-features can preserve past knowledge effectively. When the number of tasks $T$ increases, each phase introduces fewer new classes, making early-stage classification easier and causing accuracy to drop more slowly over phases; this can increase the average incremental accuracy.
>
> In addition, since all classes in CUB are birds, the concepts introduced in each phase remain broadly relevant and discriminative for bird classification, which can further improve average incremental performance when tasks are finer-grained.
>
> ### 5. Figure 6: some positive concepts disappear / diminish in later phases
>
> This behavior is expected when new, semantically related classes are introduced. In the shown example, later phases introduce other fish classes (e.g., eel and coho), which makes generic concepts that were useful early (e.g., “a fish”) less discriminative, so they drop out of the top-5 contributors. Meanwhile, more class-specific concepts (e.g., “barbel on the chin”) remain among the strongest contributors because they discriminate sturgeon from the newly added fish classes more effectively. Thus, although the set of top-5 concepts changes, the dominant surviving contributors remain semantically aligned with the target class, consistent with meaningful concept-based reasoning under increasing class complexity.
>
> More generally, it is possible that a small number of concepts have weak or unintuitive correlation with an input sample due to training dynamics and/or residual vision–language model misalignment, as discussed in Appendix A2 (Limitations). In this example, however, the contribution of such concepts is minimal, and the decision is primarily driven by the most class-relevant concepts.
>
> ### 6. Why ICICLE and IN2 are not included in later experiments (e.g., Experiment III)
>
> In Table 1, we compare against other interpretable CIL approaches (including ICICLE and IN2) under the pretrained-backbone setting, because these baselines rely on (and are reported under) a pretrained feature extractor configuration.
>
> In Tables 2 and 3, our focus shifts to non-pretrained scenarios (training the backbone from scratch in the first phase and then continuing incrementally), including both ResNet-based and DeiT-based backbones. The current interpretable baselines (ICICLE/IN2) do not support or report results under these non-pretrained protocols in a way that is directly comparable to our setups; hence we restrict Tables 2–3 to methods designed and evaluated for that regime.
>
> Finally, Table 4 uses a pretrained ViT-Base/16 setting again, where we compare against the appropriate pretrained, non-interpretable methods.
>
> ### 7. Minor comments
>
> Thank you. We addressed all minor points in the revised manuscript, including terminology, equation notation clarifications, punctuation fixes, and completing missing citations.

---

> > ### Comment · Reviewer_oTPr · 2026-02-23
> > **Re: rebuttal**
> >
> > I would like to thank the authors for their detailed rebuttal, which includes explanations and new results that have addressed my concerns. I would like to recommend the authors to include brief discussions about "4. CUB performance increases with more tasks" and "5. Figure 6: some positive concepts disappear / diminish in later phases" in their manuscript.

---

> > > ### Author Response · Authors · 2026-02-23
> > > **Added discussion on CUB trend and concept evolution**
> > >
> > > Thank you for the positive feedback and the suggestion. We have added discussions explaining the CUB trend observed in Table 1 and the evolution of the most influential concepts across phases in Fig. 6. We appreciate the reviewer’s recommendation, which helped improve the clarity of the manuscript.

---

### Review · Reviewer_jiSK · 2025-12-28

**Summary Of Contributions:**

The authors proposed CI-CBM, the extended version of Concept Bottleneck Models to exemplar-free class-incremental learning by incrementally expanding a global concept vocabulary while avoiding storage of past data.
For each phase, new concepts are generated using an LLM and aligned with images via a frozen vision–language model, and a concept bottleneck layer is trained to map visual features to concept activations.
To prevent previously learned concepts from drifting, a concept-level distillation loss enforces consistency of old concept activations across phases using only current data.
To mitigate classifier forgetting without access to old samples, the method synthesizes pseudo-features for old classes by shifting new-class features using stored class means and projects them into concept space as pseudo-concepts.
A sparse linear classifier is then trained jointly on real concepts for new classes and pseudo-concepts for old classes, enabling unified prediction across all seen classes.
At inference time, predictions rely on real concept activations, and class–concept weights provide interpretable global and local explanations.

**S1. Exemplar-Free Knowledge Retention through Pseudo-Features and Pseudo-Concepts.**
The proposed method constructively addresses the challenge of exemplar-free class-incremental learning by avoiding the need to store raw data from previous tasks.
By retaining only lightweight class-level statistics and synthesizing pseudo-features from newly observed data, the model approximates past class representations without violating memory or privacy constraints.
Projecting these pseudo-features into the concept space further enables the use of pseudo-concepts to retrain a unified classifier across old and new classes.
This strategy offers a practical and scalable alternative to exemplar replay, allowing continual adaptation while maintaining a compact memory footprint and supporting concept-based reasoning throughout training.

**S2. Comprehensive and Rigorous Experimental Evaluation.**
Another strength of the proposed method lies in its extensive and well-designed experimental evaluation.
The authors validate their approach across multiple benchmark datasets and incremental learning settings, comparing against both interpretable and non-interpretable continual learning baselines.
In addition to reporting overall accuracy and forgetting metrics, the experiments include ablation studies that isolate the contributions of key components, such as concept distillation and pseudo-feature generation.
Qualitative visualizations further illustrate how concept-based reasoning evolves across incremental phases, providing complementary insights beyond quantitative performance.
Together, these results offer a thorough assessment of both the effectiveness and behavior of the method in class-incremental scenarios.

**W1: Strong Assumptions and Backbone Dependence of Pseudo-Features.**
The pseudo-feature mechanism relies on several strong implicit assumptions, including that class-conditional feature distributions can be well approximated by their means, that nearby classes share similar covariance structures, and that feature-space proximity reflects semantic compatibility.
While these assumptions are illustrated through a toy example, they are not guaranteed to hold in complex, multimodal, or fine-grained datasets.
As a result, the separability of pseudo-features--and consequently the effectiveness of pseudo-concepts--may depend heavily on the representational quality and inductive biases of the underlying backbone.
In settings where the backbone does not produce well-structured or linearly separable feature spaces, pseudo-features may overlap or distort class boundaries, potentially limiting robustness in class-incremental learning.

**W2: Insufficient Clarity and Justification of Design Choices.**
While the overall framework is well structured, several key design choices lack sufficient clarification or justification.
In particular, the use of SigLIP for image–concept alignment is not clearly motivated, nor is its impact isolated through ablation against more standard alternatives such as CLIP.
Additionally, details of the class-incremental continual learning setup--such as phase construction, class ordering, and their interaction with concept expansion--are not always described with enough precision to fully understand or reproduce the experimental setting.
These ambiguities do not invalidate the method, but they make it harder to attribute performance gains to specific components and may limit accessibility for readers seeking to adapt or extend the approach.

**Additional Comments:**

NA

**Audience:**

Yes

**Audience Explanation:**

At least a portion of TMLR’s audience would be interested in the findings of this paper, as interpretable continual learning is an active and growing area of research at the intersection of representation learning, explainable AI, and lifelong learning.
The work addresses a timely problem--how to retain interpretability while performing exemplar-free class-incremental learning--which is relevant to researchers concerned with transparency, robustness, and deployment constraints such as memory and privacy.
By combining concept bottleneck models with continual learning mechanisms, the paper speaks to audiences working on interpretability, concept-based models, and practical continual learning systems.
As such, its methodological contributions and empirical insights are likely to be of interest to readers seeking interpretable alternatives to black-box continual learning approaches.

**Broader Impact Concerns:**

I do not identify significant broader impact concerns associated with this work.
The proposed method focuses on improving exemplar-free class-incremental learning with an emphasis on interpretability and memory efficiency, which are generally aligned with responsible and trustworthy AI development.
By avoiding storage of past data and providing concept-based explanations, the approach may even help mitigate privacy risks and improve transparency in deployed systems.
Overall, the work is methodological in nature and does not appear to introduce ethical, societal, or safety concerns beyond those commonly encountered in machine learning research.

**Claims And Evidence:**

Yes

**Claims Explanation:**

Overall, the claims made in the submission are supported by convincing, and clear empirical evidence.
The authors present thorough experimental results across multiple datasets and class-incremental settings, accompanied by ablation studies and qualitative analyses that help substantiate the effectiveness of the proposed approach.
The reported improvements over prior interpretable and exemplar-free continual learning methods are consistent and well documented, lending credibility to the main performance-related claims.

At the same time, some aspects of the evidence would benefit from additional clarification or validation.
In particular, certain design choices--such as the use of SigLIP for concept alignment and the assumptions underlying pseudo-feature generation--are not fully justified or isolated through targeted ablations.
Moreover, while the qualitative concept visualizations are informative, further analysis of the faithfulness and robustness of pseudo-concept–based reasoning would strengthen the interpretability claims.
Addressing these points would improve the clarity and completeness of the evidence, but they do not fundamentally undermine the validity of the submission’s core claims.

**Requested Changes:**

**Related to W1.**
To strengthen the robustness and interpretability claims of the proposed method, the authors are encouraged to more explicitly analyze the limitations of the pseudo-feature and pseudo-concept mechanism.

First, introducing failure cases would substantially improve the empirical analysis.
While Fig. 3 provides an intuitive toy example where pseudo-features remain well separated, it is unclear how often this assumption holds in more complex or fine-grained settings.
Visualizing or quantifying cases where pseudo-features overlap with other classes, or where pseudo-concepts poorly approximate real concept distributions, would help delineate the practical boundaries of the method.

Second, the qualitative example in Fig. 6 could be strengthened.
In its current form, many of the highlighted concepts in later phases do not appear to correspond intuitively to the class "Sturgeon", making it difficult to interpret the figure as evidence of meaningful concept-based reasoning.
Providing additional examples--ideally where the majority of contributing concepts are semantically aligned with the target class--or explicitly contrasting successful and failure cases would improve the clarity and credibility of the interpretability demonstration.

**Related to W2.**
First, the motivation for using SigLIP in the concept alignment module should be made more explicit.
Clarifying why SigLIP was preferred over more commonly used alternatives such as CLIP--particularly in the context of class-incremental learning and concept bottleneck models--would help readers better understand this design decision.
Including a simple ablation or comparison (e.g., SigLIP vs. CLIP under the same pipeline) would further strengthen attribution of performance gains and improve methodological transparency.

Second, the description of the class-incremental continual learning setup would benefit from greater precision.
Details such as how classes are partitioned into phases, how class ordering is determined, and how concept expansion interacts with phase boundaries are sometimes implicit rather than explicit.
Providing a clearer, step-by-step description of the incremental protocol--possibly with a concise table or schematic--would make the experimental setup easier to follow and reproduce.

---

> ### Author Response · Authors · 2026-02-22
> **Response to Reviewer jiSK**
>
> We thank the reviewer for the thoughtful and constructive feedback, and for recognizing the strengths of our approach and experimental evaluation. Below we respond to the requested changes point-by-point, referring to the revised manuscript where appropriate.
>
> ### 1. Limitations and backbone dependence of pseudo-features and pseudo-concepts
>
> Following the reviewer’s concern that pseudo-features may rely on strong geometric assumptions and may be sensitive to backbone quality, we added Section A15 in Appendix: Pseudo-Features Track Real Feature Geometry Under a Prototype Classifier. In this section, we empirically test whether pseudo-features preserve the same class geometry induced by real features, using a cosine-prototype (nearest-centroid) classifier.
>
> Under a CIFAR-100, $T=10$ phase class-incremental protocol, we freeze a feature extractor and compute phase-specific class centroids. We then build four “phase-by-phase” matrices (Figure A5), where entry $(i, j)$ evaluates performance on classes introduced at phase $j$ after learning up to phase $i$:
>
> - Training accuracy (cosine-prototype) on real training features,
> - Pseudo-feature accuracy (cosine-prototype) = fraction of pseudo-features whose nearest centroid is their own class centroid,
> - Test accuracy (cosine-prototype) on held-out real features, and
> - Our method’s test accuracy (not the prototype classifier).
>
> We run the same analysis under: (i) an ImageNet-pretrained ResNet-18 (matching our pretrained setting), and (ii) the same ResNet-18 trained only on first-phase data and then frozen (matching the non-pretrained regime).
>
> Across phases, the pseudo-feature matrix closely tracks the training/test matrices under the same cosine-prototype rule. In other words, when evaluated by the same nearest-centroid decision rule, pseudo-features induce similar confusion patterns / decision geometry to real features, which supports the intended role of pseudo-features as a faithful proxy for the evolving feature space.
>
> The second (first-phase-trained) backbone regime exposes the reviewer’s exact concern: classes introduced after first phase show lower prototype accuracy compared to the accuracy of first phase data simultaneously for real features and pseudo-features. A plausible explanation is that the backbone is trained only with first-phase supervision, so early classes form tighter clusters around their centroids, while later classes are less clustered and thus harder for nearest-centroid classification. Importantly, this trend is not due to pseudo-features alone—real features and pseudo-features drop together under the same cosine-prototype rule.
>
> Despite this representation constraint, our full method still improves test accuracy in both the ImageNet-pretrained and the first-phase-trained settings, with particularly larger gains in the latter. This suggests that the concept bottleneck provides an additional structured space that increases separability even when the raw feature space is not well organized for newly introduced classes. Beyond accuracy, this also strengthens interpretability: predictions remain grounded in a small set of activated concepts rather than relying solely on fragile feature-space nearest-centroid structure.

---

> ### Author Response · Authors · 2026-02-22
> **Response to Reviewer jiSK (Part 2)**
>
> ### 2. Strengthening the qualitative interpretability example in Fig. 6 (and pointing to additional visualizations in A16)
>
> Regarding the reviewer’s comment that the later-phase top concepts for “Sturgeon” may look less intuitive, we clarify that this example intentionally illustrates how concept contributions re-rank as new confusable classes appear.
>
> In this particular run, in Phase 2 and Phase 3, other fish classes (e.g., Eel and Coho) are introduced. As a result, some generic concepts that were highly predictive in Phase 1 (e.g., “a fish”) naturally drop out of the top-5 contributions, because they are no longer discriminative among multiple fish categories. Meanwhile, a more class-discriminative concept (e.g., “barbel on the chin”) remains among the strongest contributors, since it separates Sturgeon from the newly introduced fish classes more effectively. Thus, although the set of top-5 concepts changes, the dominant surviving contributors remain semantically aligned with the target class, consistent with meaningful concept-based reasoning under increasing class complexity.
>
> More generally, we note that it is possible for a small number of concepts to have weak or unintuitive correlation with a given input sample due to the training dynamics and vision–language model misalignment, as discussed in Appendix A2 (Limitations). Importantly, this observation is not specific to Fig. 6: it is a general limitation of concept attribution pipelines that rely on vision–language supervision. In the Sturgeon example, however, the contribution of such concepts is minimal, and the prediction is primarily driven by the most class-relevant and discriminative concepts (e.g., “barbel on the chin”), which remain stable among the highest contributors even after other fish classes are introduced.
>
> In addition, we emphasize that Appendix A16 (Additional Visualization for Model Reasoning) was already included in the original manuscript and provides further qualitative evidence for both local and global interpretability across additional classes and datasets.
>
> ### 3. Motivation for using SigLIP and SigLIP vs. CLIP ablation
>
> Following the reviewer’s suggestion, we made this design choice explicit and isolated its impact via a controlled ablation in the revised manuscript (Appendix A14: Impact of Vision-Language Model (SigLIP vs. CLIP)). In this study, we keep the entire training pipeline identical and only swap the vision–language model used to compute the concept-activation matrix $P$ (CLIP vs. SigLIP). The results show that using SigLIP yields consistently higher class-incremental accuracy across datasets and phase configurations, indicating stronger image–concept alignment in $P$ and better downstream performance.
>
> | Method | CIFAR-100 (T=5) | CIFAR-100 (T=10) | CIFAR-100 (T=20) | CUB (T=4) | CUB (T=10) | CUB (T=20) | TinyImageNet (T=5) | TinyImageNet (T=10) | TinyImageNet (T=20) |
> |---|---:|---:|---:|---:|---:|---:|---:|---:|---:|
> | CI-CBM (CLIP) | 68.6 | 68.7 | 67.2 | 60.5 | 62.4 | 63.6 | 47.0 | 47.2 | 47.3 |
> | CI-CBM (SigLIP) | **68.8** | **68.8** | **67.8** | **62.2** | **65.3** | **66.1** | **48.6** | **48.7** | **48.5** |
>
> In addition, we evaluate the effect on concept fidelity, whether the concept assigned to a bottleneck neuron remains faithful to the same semantic concept at the end of continual training, using the protocol and metrics defined in Appendix A13: Impact of the Distillation Regularizer on Concept Fidelity. Under the same controlled swap, SigLIP improves concept-fidelity scores in all settings, which strengthens the methodological justification beyond accuracy alone.
>
> | Metric | VLM | CIFAR100 (T=5) | CIFAR100 (T=10) | CIFAR100 (T=20) | CUB (T=4) | CUB (T=10) | CUB (T=20) | TinyImageNet (T=5) | TinyImageNet (T=10) | TinyImageNet (T=20) |
> |---|:---:|---:|---:|---:|---:|---:|---:|---:|---:|---:|
> | CLIP cosine similarity | CLIP | 0.890 | 0.888 | 0.890 | 0.949 | 0.854 | 0.830 | 0.866 | 0.866 | 0.845 |
> | CLIP cosine similarity | SigLIP | **0.927** | **0.929** | **0.837** | **0.961** | **0.876** | **0.837** | **0.883** | **0.886** | **0.869** |
> | Top-5 accuracy | CLIP | 73.6 | 74.9 | 49.4 | 72.9 | 58.5 | 53.4 | 66.6 | 66.5 | 57.0 |
> | Top-5 accuracy | SigLIP | **87.7** | **87.0** | **71.8** | **78.9** | **68.8** | **49.4** | **74.7** | **74.0** | **66.5** |

---

> ### Author Response · Authors · 2026-02-22
> **Response to Reviewer jiSK (Part 3)**
>
> ### 4. Greater precision on the class-incremental setup (class ordering, phase construction, and interaction with concept expansion)
>
> In the revised manuscript, we have strengthened the description of (a) class ordering, (b) phase construction, and (c) how these choices interact with our concept-set expansion, with the goal of making the experimental setting fully reproducible.
>
> **Class ordering and phase construction.**
> In Section 4 (Evaluation), paragraph “Datasets and Implementation Details”, we already state that, for fair comparison, we adopt the same procedure as prior CIL baselines by shuffling the class order using the same random seed as the compared methods and then splitting the shuffled list into phases. To remove any ambiguity, we have now explicitly added the exact seed value (1993) used to generate the class order and phase splits.
>
> **Explicit phase configurations for each experimental group.**
> We further clarified phase construction by explicitly tying each evaluation group to its phase split definition:
>
> - For Experiment I (Table 1), we already state in “Comparison to Interpretable Methods” that classes are distributed evenly across $T$ phases; we emphasize this sentence as the governing protocol for Table 1.
> - For Experiments II–IV (Table 2, Figure 5, Table 3), we provide the non-uniform class splits (initial base classes followed by incremental phases) in the second paragraph of “Comparison to Non-Pretrained and Non-Interpretable Methods,” where we explicitly state the class-splitting protocol adopted from prior work.
> - For the pretrained, non-interpretable setting (Experiment V), we again use even class distribution across $T$ phases, and we added a short clarifying sentence to make explicit that the pretrained-backbone experiments follow this same even-split protocol.
>
> **(c) Interaction with concept expansion.**
> Our concept-set expansion is performed per phase based on the newly introduced classes, and the resulting concept set is cumulative across phases. To make the filtering/uniqueness criteria fully transparent, we added an explicit clarification in Module (I) (Incremental Concept Set Expansion) and pointed readers to the full pipeline in Appendix A11 (Concept Generation Pipeline). Concretely, we now specify that we embed candidate concepts, existing concepts, and the names of seen classes with text encoders (Sentence-Transformer and the SigLIP text encoder), compute cosine similarities, and filter candidates using fixed thresholds to avoid redundancy and class-name leakage; the full generation and filtering procedure is provided in Appendix A11.

---

> > ### Comment · Reviewer_jiSK · 2026-03-09
> > **Re:rebuttal**
> >
> > I would like to thank the authors for their detailed rebuttal and the effort they put into addressing many of my concerns.
> > I would also like to point out a few minor typos in A15:
> > - The text "(i) an ImageNet-pretrained ResNet-18 (corresponding to Experiment I in Table 1)" appears to be repeated.
> > - In the third bullet point, the word "Test" is missing the letter "T."

---

> > > ### Author Response · Authors · 2026-03-09
> > > **Response to Reviewer Comment on Appendix A15**
> > >
> > > Thank you for the careful review and for pointing out these typos. We have corrected both issues in Appendix A15. The revised version with these corrections has been uploaded.

---

### Review · Reviewer_urji · 2026-02-09

**Summary Of Contributions:**

CI-CBM is an extension of Concept Bottleneck Models in the exemplar-free class incremental learning setting. The goal is to maintain interpretability but also mitigate catastrophic forgetting. The proposed method incrementally expands a set of human-interpretable concepts generated by LLMs and aligned to visual features via CLIP/SigLIP. CI-CBM incorporates a distillation-based regularization term on the concept bottleneck layer to preserve old concepts as we learn new concepts. Since data from earlier classes is unavailable, the method adopts a pseudo-feature generation strategy (based on feature-space translation from newly observed classes) to construct pseudo-concepts for past classes. Those pseudo concepts are then used together with current data to train a sparse, unified classifier across all phases. The paper provides a geometric intuition for the pseudo feature mechanism and evaluates CI-CBM on several standard class incremental benchmarks under both pre-trained and non-pretrained backbones. The experiments show improvements over prior interpretable CIL methods and similar performance relative to non-interpretable approaches.

**Additional Comments:**

see above

**Audience:**

Yes

**Audience Explanation:**

I have no doubt about this. Researchers interested in interpretable machine learning, concept bottleneck models, and continual or class-incremental learning would find the paper of interest. I am sure that the communities that work on exemplar-free continual learning, interpretability by design, or the integration of language- and vision-based representations will appreciate the paper’s empirical investigation.

**Broader Impact Concerns:**

the work does not raise any ethical concerns that would require an extensive Broader Impact discussion.

**Claims And Evidence:**

Yes

**Claims Explanation:**

I answered "yes" because  there is no "partially" option. The empirical results are clear and they support the claim that CI-CBM outperforms prior interpretable CIL methods on the given benchmarks and maintains a degree of concept-level interpretability across phases. The ablation studies also provide evidence that key components of the method (such as pseudo-concept generation and concept regularization) really contribute to performance.

However several of the broader claims are only weakly supported. Some conclusions are drawn from a specific experimental regime with particular design choices (eg, frozen backbones, centroid-based pseudo-feature translation, CLIP-aligned concepts, sparse linear classifiers). It is unclear to me how robust these findings are beyond these conditions. In addition some claims about general applicability, stability of learned concepts and the ability to “maintain human-understandable concepts” throughout incremental learning rely primarily on visualizations and limited experiments. I would like to see more systematic or task-agnostic measures of interpretability and concept fidelity. In conclusion the paper is less convincing in supporting the stronger, more general claims made about interpretability and continual learning behavior in broader settings.

**Requested Changes:**

The following three papers are very relevant in my opinion and should be cited and contrasted with CI-CBM:

FeTrIL++ https://arxiv.org/abs/2403.07406
CI-CBM’s pseudo-concept generation mechanism is mathematically identical to FeTrIL-style pseudo-feature translation, right? It seems to me that the only difference is that pseudo-features are projected into a concept space. FeTrIL++ provides a much deeper analysis of when centroid-shift-based pseudo-features work or fail.

CONCIL: https://arxiv.org/abs/2411.17471
this paper defines the concept and class incremental learning problem for CBMs as a first-class task and proposes a different solution based on analytic (non-gradient) updates with theoretical guarantees of no forgetting. CI-CBM tackles a similar problem but with heuristic, gradient-based mechanisms. The paper should clarify how its goals and scope differ from CONCIL. I suggest that the authors also temper their claims that CI-CBM represents a very principled solution to CBMs under continual learning

Language-Guided Concept Bottleneck Models for Interpretable Continual Learning:
https://openaccess.thecvf.com/content/CVPR2025/papers/Yu_Language_Guided_Concept_Bottleneck_Models_for_Interpretable_Continual_Learning_CVPR_2025_paper.pdf
This paper already combines LLM-generated concepts, CLIP alignment, semantic consistency losses, and prototype-based pseudo-replay for exemplar-free class-incremental learning. CI-CBM follows a very similar design pattern. The authors should explain more clearly what is genuinely new relative to this paper. I don't really see the major novelty that CI-CBM brings in this line of work.

Some more specific comments:


The distillation-style loss on the concept bottleneck layer is central but its effect is not well isolated. can you show how concept-level accuracy or stability changes when this regularization is removed? (I mean beyond just showing downstream classification accuracy)


The pseudo-feature translation is taken from FeTrIL. Plz discuss: why translating features and then projecting to concepts is preferable to translating concept activations directly. Are the two equivalent under certain assumptions?


The Gaussian toy example assumes matched or near-matched variances to obtain linear decision boundaries. This assumption is strong and mirrors assumptions in FeTrIL++


The comparison between GPT-generated concepts and ConceptNet concepts is useful. However, the failure on CUB suggests that concept quality is a critical bottleneck. This deserves more emphasis in the main paper as it directly affects the robustness of the approach.


The paper shows that performance degrades gracefully when fewer concepts are available. It would be useful to connect this result to interpretability: does reduced concept availability also reduce explanation quality, or mainly accuracy?


The “unique concept expansion” mechanism is interesting but its impact is mostly discussed in terms of model size. A brief analysis of how duplicate concepts affect interpretability (not just efficiency) would strengthen this section.

Results in the one-class-increment regime are much weaker than full rehearsal. This setting highlights limitations of pseudo-concept generation and deserves more discussion because it is a realistic EF-CIL scenario.

---

> ### Author Response · Authors · 2026-02-22
> **Response to Reviewer urji**
>
> We thank the reviewer for the detailed feedback and for recognizing that our empirical results and ablations support the core claims. Below we respond point-by-point and summarize the corresponding updates in the revised manuscript.
>
> ### 1. Scope of claims and robustness beyond the experimental regime
>
> **Frozen backbone and centroid-based pseudo-feature translation.**
> The centroid-shift pseudo-feature translation uses stored class centroids as anchors, so it benefits from a feature space that stays comparable across phases. If the backbone continues to fine-tune, the representations of previously seen classes can drift, making earlier centroids stale; updating them would typically require access to past data, which is unavailable in exemplar-free class-incremental learning. Accordingly, we freeze the backbone as a natural choice in this setting, since it keeps stored centroids meaningful without needing old data to re-estimate them.
>
> **Sparse linear classifier choice.**
> We also ablated the sparsity constraint in Appendix A5 / Table A5 (Experiment XI: Impact of Sparsity on Model Performance) in the original submission. Removing sparsity does not improve performance. In the revision, we additionally include a new decision-making visualization to more directly assess how a dense classifier allocates concept-level evidence, using the same per-image concept contribution analysis as our sparse predictor. Specifically, Figure A1 (Appendix A5) repeats this analysis but trains the final predictor without the sparsity regularizer. Although the dense model still classifies the example correctly, the attribution mass is spread across many concepts: the top-ranked concepts each contribute only marginally to the predicted class, while the aggregate contribution of the remaining concepts dominates. This yields explanations that are less concise and harder to interpret, since there is no small set of core concepts with clearly dominant contributions.
>
> **More systematic concept fidelity / stability evidence.**
> We added explicit concept-fidelity evaluations in the revision that quantify semantic stability of bottleneck units over incremental phases. Concretely, on the test split we first build a concept-activation matrix $P$ using a vision–language model, where each entry measures image–text alignment between a test image and a concept name. We then record each bottleneck unit’s activation vector over the same test images and assign a concept label to each unit by selecting the concept whose $P$-profile is most cosine-similar to that unit’s activation profile. Using this assignment, we report two complementary fidelity metrics: (i) CLIP-space cosine similarity between predicted vs. reference concept text embeddings, and (ii) Top-5 concept accuracy (whether the reference concept appears among the five closest concepts).
>
> In Appendix A13, we isolate the effect of the bottleneck distillation regularizer by comparing concept-fidelity scores with vs. without the regularizer under the same incremental protocols and across different phase granularities $T$. We find the regularizer consistently improves these fidelity measures across datasets, indicating reduced semantic drift of concept units during continual learning.
>
> In Appendix A14, we test robustness to the vision–language alignment choice by computing $P$ with SigLIP vs. CLIP (keeping the training protocol fixed), and observe that SigLIP generally yields higher concept-fidelity scores (and often slightly better accuracy), suggesting improved semantic alignment of learned concept units when the underlying VLM provides stronger image–text alignment.

---

> ### Author Response · Authors · 2026-02-22
> **Response to Reviewer urji (Part 2)**
>
> ### 2. Requested related work and novelty clarification
>
> We agree these papers are relevant. We added them to the Related Work and discussed them in the appropriate experiment sections, and we also incorporated additional baselines as suggested.
>
> **FeTrIL++ and relation to our pseudo-feature mechanism.**
> Yes. Our pseudo-feature translation is closely related in spirit to FeTrIL-style centroid shift. Our key difference is how pseudo-samples are used: CI-CBM maps pseudo-features through the learned concept bottleneck to produce pseudo-concepts, enabling a unified concept-based classifier and consistent concept-level explanations across phases. We also acknowledge that FeTrIL++ provides a deeper discussion of when centroid-based pseudo-replay succeeds or fails. We add complementary empirical observations in Appendix A15 of the revised manuscript.
>
> **CONCIL: goals/scope differences and claims.**
> We added and contrasted CONCIL in Related Work and experiments. While CONCIL addresses a related continual CBM objective, it differs substantially in assumptions and scope. In particular, CONCIL requires per-sample concept annotations, which can be impractical in many real-world scenarios where concept labels are not available at scale. Additionally, its reported evaluation is limited to the CUB dataset, while CI-CBM evaluates across multiple standard EF-CIL benchmarks and settings. We also added CONCIL’s results to Table 1 (Experiment I), where CI-CBM outperforms it under the same class-incremental evaluation protocol. Finally, we revised the manuscript wording to avoid suggesting that CI-CBM provides theoretical no-forgetting guarantees or constitutes an analytic solution.
>
> **Language-Guided CBM for Interpretable Continual Learning (CVPR 2025) and novelty.**
> We added this paper (Language-Guided CBM) to Related Work and added its results to Table 4 (Experiment V). Under the same evaluation setting, CI-CBM outperforms it on both CIFAR-100 and CUB. Methodologically, while both approaches use LLM-generated concepts and VLM alignment, CI-CBM differs from Language-Guided CBM in two key aspects:
>
> 1) **Bottleneck distillation targeted at concept fidelity.**
>    CI-CBM introduces a dedicated bottleneck distillation objective to preserve concept meaning across phases. This component is not present in Language-Guided CBM. Empirically, Appendix A13 shows it improves concept stability across phases, and Sections A12–A13 demonstrate consistent gains in both performance and interpretability by mitigating semantic drift in bottleneck units.
>
> 2) **Reduced reliance on large-scale pretraining.**
>    Language-Guided CBM relies on a large-scale pretrained CLIP backbone. Since a core goal of CIL is to acquire knowledge that was previously unavailable to the model, heavy pretraining raises a practical question about whether the model truly encounters novel information. To address this directly, we report results in both pretrained and non-pretrained settings, showing CI-CBM remains effective even when the backbone is not heavily pretrained, which we argue better reflects realistic deployment scenarios.

---

> > ### Author Response · Authors · 2026-02-22
> > **Response to Reviewer urji (Part 3)**
> >
> > ### 3. Distillation loss: isolating concept stability beyond downstream accuracy
> >
> > We agree that downstream accuracy alone is insufficient to justify the distillation regularizer. This is why we added Appendix A13: Impact of the Distillation Regularizer on Concept Fidelity, which directly measures concept stability/semantic drift.
> >
> > | Metric | VLM | CIFAR100 (T=5) | CIFAR100 (T=10) | CIFAR100 (T=20) | CUB (T=4) | CUB (T=10) | CUB (T=20) | TinyImageNet (T=5) | TinyImageNet (T=10) | TinyImageNet (T=20) |
> > |---|:---:|---:|---:|---:|---:|---:|---:|---:|---:|---:|
> > | CLIP cosine similarity | CLIP | 0.890 | 0.888 | 0.890 | 0.949 | 0.854 | 0.830 | 0.866 | 0.866 | 0.845 |
> > | CLIP cosine similarity | SigLIP | **0.927** | **0.929** | **0.837** | **0.961** | **0.876** | **0.837** | **0.883** | **0.886** | **0.869** |
> > | Top-5 accuracy | CLIP | 73.6 | 74.9 | 49.4 | 72.9 | 58.5 | 53.4 | 66.6 | 66.5 | 57.0 |
> > | Top-5 accuracy | SigLIP | **87.7** | **87.0** | **71.8** | **78.9** | **68.8** | **49.4** | **74.7** | **74.0** | **66.5** |
> >
> > We track each bottleneck neuron’s assigned concept over phases and evaluate whether the neuron remains faithful to the same semantic concept at the end of continual learning. We quantify fidelity using two complementary metrics:
> > i) CLIP-space cosine similarity between the predicted concept label and the ground-truth assigned concept, and
> > ii) Top-5 concept accuracy (whether the ground-truth concept remains among the five most similar concepts by cosine similarity).
> >
> > Across datasets and phase granularities, enabling the distillation regularizer consistently improves both concept-fidelity metrics, supporting the claim that distillation mitigates semantic drift and helps preserve stable, human-interpretable concepts throughout incremental learning. We also added Appendix A14 to show that the choice of VLM (SigLIP vs. CLIP) affects concept fidelity as well, further supporting that interpretability depends on both the regularizer and the quality of the image–concept alignment.
> >
> > ### 4. Why translate features then project to concepts
> >
> > We addressed this question directly via a new ablation in Section A8 “Alternative Approach for Learning the Prediction Layer” / Table A8.
> >
> > We test a natural concept-space variant: we translate class prototypes directly in concept space. Concretely, we form a concept-space centroid for each past class using the bottleneck representation before updating on the current phase; because new concept dimensions do not exist in earlier phases, we pad missing dimensions with zeros and then apply the same prototype-translation rule in concept space.
> >
> > | Method | CIFAR-100 (T=5) | CIFAR-100 (T=10) | CIFAR-100 (T=20) | TinyImageNet (T=5) | TinyImageNet (T=10) | TinyImageNet (T=20) |
> > |---|---:|---:|---:|---:|---:|---:|
> > | Concept-Space Prototype Generation | 48.2 | 35.8 | 27.2 | 33.4 | 25.3 | 18.9 |
> > | CI-CBM | **68.8** | **68.8** | **67.8** | **48.6** | **48.7** | **48.5** |
> >
> > This concept-space translation performs consistently worse across settings. The main reason is that concept space is not stationary across phases: the bottleneck both expands (new concept coordinates are added) and can drift due to continued alignment/distillation, so past class centroids become misaligned with the updated concept basis. In contrast, the backbone feature space is stable (frozen in our setting), and projecting translated pseudo-features through the current bottleneck automatically expresses pseudo-samples in the current concept basis, yielding more robust performance.

---

> > > ### Author Response · Authors · 2026-02-22
> > > **Response to Reviewer urji (Part 4)**
> > >
> > > ### 5. Gaussian toy example assumptions and connection to FeTrIL++
> > >
> > > We agree that the Gaussian toy example relies on simplified conditions (e.g., near-matched variances) to yield clean linear decision boundaries, and is meant primarily as geometric intuition. To complement this idealized setting and assess whether our pseudo-feature translation behaves similarly under realistic, multi-class feature distributions, we add a controlled diagnostic in Appendix A15 (“Pseudo-Features Track Real Feature Geometry Under a Prototype Classifier”).
> > >
> > > The goal is to test whether pseudo-features induce decision geometry similar to real samples in a realistic multi-class EF-CIL setting. Concretely, under a $T=10$ class-incremental protocol on CIFAR-100 with a frozen feature extractor, we evaluate a simple cosine-prototype classifier built from per-class centroids, and compare phase-by-phase confusion/accuracy patterns for: (i) real training features, (ii) pseudo-features generated for past classes, and (iii) held-out real test features (all evaluated using the same nearest-centroid decision rule).
> > >
> > > Across phases, we find that the pseudo-feature results closely track the corresponding real-feature behavior: pseudo-features yield phase-by-phase patterns that are highly consistent with those produced by real training and test samples under the same prototype classifier. This indicates that pseudo-features approximate the effective class geometry relevant to centroid-based decisions.
> > >
> > > We repeat the same analysis under two backbone regimes: (i) an ImageNet-pretrained ResNet-18 and (ii) a non-pretrained setting where the backbone is trained only on the first-phase data and then frozen. In the first-phase-trained setting, prototype accuracy is systematically higher for early-phase classes and lower for later-phase classes across all three real/pseudo matrices, consistent with a representation-quality limitation (later classes are less well clustered around a single centroid because they did not participate in backbone training). Importantly, this degradation is not specific to pseudo-features: it affects real and pseudo features similarly, highlighting that centroid-based replay quality depends on how well the frozen feature space clusters newly introduced classes.
> > >
> > > Finally, we connect this diagnostic back to CI-CBM: our full method achieves higher test accuracy than the prototype baseline, with particularly pronounced gains in the non-pretrained setting. This supports our claim that the concept bottleneck provides an additional structured space that improves separability beyond raw centroid geometry, while also enabling consistent concept-based explanations across phases.
> > >
> > > ### 6. Concept quality as a bottleneck and the CUB/ConceptNet result
> > >
> > > We agree that concept quality is crucial: irrelevant or low-coverage concepts can hurt both accuracy and interpretability. For the specific CUB/ConceptNet failure, the issue is primarily concept availability and coverage: ConceptNet provides too few associated concepts for many fine-grained CUB classes, and the resulting concept pool is much smaller than what GPT-generated concepts can provide.
> > >
> > > We support this explanation with Table A6 (Experiment XII: Accuracy vs. concept availability): reducing concepts moderately causes only a small accuracy drop in general, but in the CUB+ConceptNet case the concept pool becomes too small, which makes the bottleneck under-specified and harms performance. In addition, Table A7 (Accuracy vs. SNR in image–concept alignment) shows that the method is relatively insensitive to moderate noise in the concept activation matrix $P$: accuracy degrades only marginally as SNR decreases. Taken together, these results suggest that the main failure mode on CUB+ConceptNet is not noise in concept supervision, but rather insufficient concept coverage.

---

> > > > ### Author Response · Authors · 2026-02-22
> > > > **Response to Reviewer urji (Part 5)**
> > > >
> > > > ### 7. The effect of reduced concept availability on interpretability
> > > >
> > > > In the revision, we added two qualitative analyses (Figure A2 and Figure A3 in Section A6 Effect of Concept Set Size on Performance). We repeat our interpretability evaluations under a reduced-concept setting where 50% of concepts are randomly masked. At the class level, Figure A2 shows that the final-layer weight structure for Tree Swallow on CUB remains coherent across phases: when fine-grained class-defining concepts are unavailable, the model shifts to semantically/visually related alternatives while continuing to accumulate informative negative (NOT) concepts. At the instance level, Figure A3 shows that concept attributions for the same Sturgeon example remain visually aligned with the input and follow a similar reasoning pattern to the full-concept setting.
> > > >
> > > > Overall, under 50% concept masking, CI-CBM preserves both global and local interpretability. When fine-grained class-defining concepts are missing, explanations rely on closely related visual alternatives; interpretability is expected to degrade more noticeably only under much smaller concept pools where such attributes are consistently unavailable.
> > > >
> > > > ### 8. Unique concept expansion: interpretability impact beyond efficiency
> > > >
> > > > We agree that duplicates matter not only for efficiency but also for interpretability. If the same concept appears multiple times in the concept set, it can fragment attribution across duplicate dimensions and make explanations less concise (even if the semantic content is similar). Removing duplicates (keeping only one instance) does not reduce the model’s ability to use that semantic cue; instead it makes the explanation cleaner by concentrating weight and attribution onto a single concept dimension. To address this point, we have added a brief discussion in Section A9 Unique Concept Expansion.
> > > >
> > > > ### 9. One-class-increment regime vs full rehearsal
> > > >
> > > > We agree this regime is challenging and realistic for EF-CIL. In the one-class-increment setting, each phase adds only one class, so early concepts and decision boundaries must remain stable across many phases while the model continues to incorporate small, incremental changes. This increases overlap and makes pseudo-replay intrinsically harder, since small updates accumulate and the model must correctly classify early classes across a long sequence of phases without access to old data.
> > > >
> > > > It is also expected that full rehearsal performs better because it has access to all previous data and can update representations and class statistics directly. However, we emphasize that CI-CBM’s gap to full rehearsal in this extremely strict setting is moderate given that CI-CBM still achieves competitive performance while maintaining concept-based interpretability.

---

### Author Response · Authors · 2026-02-22
**Global Response: Summary of Rebuttal**

We thank all reviewers for their constructive comments and for highlighting the strengths of our submission, including its timely and important problem setting (Reviewer oTPr, urji, and jiSK), clear and convincing empirical evidence (Reviewer urji, jiSK, and oTPr), and comprehensive, extensive, and well-designed experimental evaluation (Reviewer jiSK and urji). We also appreciate the recognition that the paper is well-written and easy-to-follow (Reviewer oTPr), that the proposed approach is novel in how it brings together complementary ideas into an interpretable exemplar-free CIL framework (Reviewer oTPr), and that it achieves significant and consistent gains over prior interpretable baselines while remaining competitive with non-interpretable methods (Reviewer oTPr, urji, and jiSK).

In the revised manuscript (all additions/edits highlighted in blue), we strengthened positioning and evidence in three main ways:

(i) **Expanded related work and comparisons.** We now explicitly discuss other interpretable exemplar-free CIL/CBM methods (e.g., CONCIL and CLG-CBM) and clarify connections to feature-translation approaches such as FeTrIL++.

(ii) **Improved method clarity and reproducibility.** We added implementation and protocol details, including Algorithm 1, and expanded our description of the concept deduplication filter used to avoid redundant concepts during expansion.

(iii) **Added targeted new ablations and analyses.** We include: interpretability without sparsity (Fig. A1); interpretability under reduced concept availability (Figs. A2--A3); an alternative pseudo-sample variant showing concept-space prototype translation is unreliable (Table A8); sensitivity of performance to the distillation weight (Table A10); concept-fidelity metrics validating the distillation regularizer (Table A11); and SigLIP vs. CLIP effects on both accuracy and concept fidelity (Tables A12--A13).

---

> ### Comment · Reviewer_urji · 2026-03-08
>
> I would like to thank the authors for being thorough and professional in their rebuttal and in revising the paper. My concerns have been addressed, at least in the best way the authors can realistically do so. I recommend the acceptance of this paper.

---

> > ### Author Response · Authors · 2026-03-09
> > **Response to Reviewer urji**
> >
> > We thank the reviewer for the careful evaluation of our revision and for the positive recommendation. We sincerely appreciate the time and effort invested in reviewing our work.

---

> ### Comment · Reviewer_jiSK · 2026-03-11
> **Official Comment by Reviewer jiSK**
>
> I would like to thank the authors for their thorough rebuttal and revisions to the paper.
> My concerns have been addressed to the extent reasonably possible.
> Thus, I recommend acceptance of this paper.

---

> > ### Author Response · Authors · 2026-03-11
> > **Response to Reviewer jiSK**
> >
> > We sincerely thank the reviewer for the careful evaluation of our revision and for the positive recommendation. We greatly appreciate the time and effort invested in reviewing our work and are grateful that our rebuttal and revisions were helpful in addressing the concerns.

---

### Decision · Action_Editor_pqrt · 2026-03-23

**Recommendation:** Accept as is

**Additional Comments:**

The authors clarified on the novelty w.r.t. the related work, provided new ablation studies, improved analysis and clarified upon the methodology. The reviewers all converged that the comments have been sufficiently addressed, so the paper can now be accepted.

**Audience:**

Yes

**Audience Explanation:**

I believe that the paper has an audience in interpretable machine learning, concept bottleneck, and class-incremental learning communities.

**Claims And Evidence:**

Yes

**Claims Explanation:**

This work proposes CI-CBM, an extension of Concept-Bottleneck models for class-incremental learning. This work, therefore, is at the intersection of interpretable and incremental machine learning. The authors use LLM-generated and CLIP-aligned concepts, and define a distillation-based regularisation term, which allows to preserve old concepts when learning new ones. To  this end the authors introduce the idea of pseudo-concepts for the previous classes, which are used jointly with the current data to train a sparse joint classifier for both old and new classes. The provided experiments show improvements over prior interpretable contunual-learning methods and are comparable in their performance to existing non-interpretable methods.

While initially the reviewers had a number of concerns related to the experiments and design choices, this has been addressed in detail by the authors in the rebuttal. All the reviewers are equivocal is supporting that the paper is supported by accurate, convincing and clear evidence.